# REVISITING THEORY OF CONTRASTIVE LEARNING FOR DOMAIN GENERALIZATION

## ABSTRACT

Contrastive learning is among the most popular and powerful approaches for self-supervised representation learning, where the goal is to map semantically similar samples close together while separating dissimilar ones in the latent space. Existing theoretical methods assume that downstream task classes are drawn from the same latent class distribution used during the pretraining phase. However, in real-world settings, downstream tasks may not only exhibit *distributional shifts* within the same label space but also introduce new or broader label spaces, leading to *domain generalization* challenges. In this work, we introduce *novel generalization bounds* that explicitly account for both types of mismatch: domain shift and domain generalization. Specifically, we analyze scenarios where downstream tasks either (i) draw classes from the same latent class space but with shifted distributions, or (ii) involve new label spaces beyond those seen during pretraining. Our analysis reveals how the performance of contrastively learned representations depends on the statistical discrepancy between pretraining and downstream distributions. This extended perspective allows us to derive provable guarantees on the performance of learned representations on average classification tasks involving class distributions outside the pretraining latent class set.

## 1 INTRODUCTION

Contrastive learning has emerged as one of the most effective approaches to self-supervised and unsupervised representation learning, achieving state-of-the-art results across a wide range of domains, including natural language processing (NLP; Devlin et al. (2018); Brown et al. (2020); Saggau et al. (2023)), computer vision(Chen et al., 2020; Bardes et al., 2021; Grill et al., 2020; Rezaei et al., 2023a), multimodal learning (Radford et al., 2021; Li et al., 2022; Shi et al., 2022), and bioinformatics (Gündüz et al., 2023; Rezaei et al., 2023b).

From a theoretical standpoint, a prominent framework introduced by Saunshi et al. (2019) models contrastive learning via a latent class assumption, where semantically similar data points are independently drawn from the same unobserved class. This framework associates the minimization of unsupervised contrastive loss to provable guarantees on downstream classification performance, particularly via the mean classifier. Building upon this foundation, subsequent theoretical studies have extended the analysis to domain-specific scenarios (Ko et al., 2022) and task-specific settings (Van Gansbeke et al., 2021; Shen et al., 2022).

A primary limitation of current theoretical studies is the strong assumption that the downstream data distribution matches the latent class distribution encountered during pretraining. However, in practice, downstream data may exhibit distribution shifts, where the class-conditional distributions at test time differ from those seen during pretraining. Such shifts — encompassing covariate shift, label shift, or semantic mismatch — are central to the challenge of domain generalization, and they critically affect the transferability of contrastively learned representations.

We provide a theoretical analysis of contrastive learning under distribution shift and domain generalization, extending the mean classifier framework to explicitly incorporate these aspects. We formalize distribution shift as deviations between the mean representations of pretraining and downstream class distributions, and domain generalization as the inclusion of new or expanded label spaces beyond those observed during pretraining. Building on this, we derive generalization bounds for the downstream supervised loss that include an explicit bias term induced by these mismatches.

This bias quantifies how representational misalignment between pretraining and downstream tasks impacts generalization performance. Furthermore, we show that the bias can be tightly bounded under assumptions such as Lipschitz continuity of the loss function, norm-bounded or sub-Gaussian encoders, and structured class similarities. Our framework thus unifies representation robustness analysis with both shifted and novel task domains.

To summarize, our contributions are: (i) We present a theoretical framework for contrastive learning under both domain shift and domain generalization by extending the classical latent class model of Saunshi et al. (2019) to account for mismatches between pretraining and downstream data distributions. (ii) We derive generalization bounds for downstream supervised loss that incorporate both types of mismatch, introducing an explicit bias term quantifying representational misalignment and variance terms capturing within-class variability. (iii) We show how the bias term can be bounded under realistic assumptions on the loss function (e.g., Lipschitz continuity) and encoder properties (e.g., norm-boundedness, sub-Gaussian concentration), covering a broad family of encoders used in practice.

## 2 BACKGROUND

**Contrastive learning** The goal of contrastive learning is to learn a representation space where semantically similar samples are close, and dissimilar ones are far apart. Let $f_\theta \in \mathcal{F}$ denote an encoder with parameters $\theta$, mapping inputs $x \in \mathcal{X}$ to embeddings $z = f_\theta(x) \in \mathbb{R}^d$. We assume bounded embeddings, $\|f_\theta(x)\| \leq R$. Training requires a pair of similar samples $(x, x^+)$ drawn from the same distribution $D_{\text{sim}}$, often implemented via augmentations of the same example. Negative samples $x_1^-, \ldots, x_k^-$ are typically taken as the other samples in the minibatch that are not part of the positive pair $D_{\text{neg}}$. Under the latent class model (Saunshi et al., 2019), this is abstracted as negatives being drawn from the marginal distribution $\mathcal{D}_{\text{marg}} = \mathbb{E}_{c \sim \rho}[\mathcal{D}_c]$.

Given a minibatch of $N$ positive pairs, we denote $z_j = f_\theta(x_j)$. The standard NT-Xent (InfoNCE) loss for a positive sample $(i, j)$ is computed by:

$$\ell(i, j) = -\log \frac{\exp\left(\text{sim}(z_i, z_j)/\tau\right)}{\sum_{m=1}^{2N} \exp\left(\text{sim}(z_i, z_m)/\tau\right)}, \tag{1}$$

where $\text{sim}(\cdot, \cdot)$ is cosine similarity and $\tau > 0$ is a temperature parameter.

The contrastive loss aggregates over all samples:

$$\mathcal{L}_{\text{contrastive}} = \frac{1}{2N} \sum_{m=1}^{N} \left[ \ell(2m - 1, 2m) + \ell(2m, 2m - 1) \right]. \tag{2}$$

Following Saunshi et al. (2019), this objective can be viewed as minimizing the unsupervised population loss

$$\mathcal{L}_{\text{un}}(f) := \mathbb{E}\left[ \ell\left(\{f(x)^\top(f(x^+) - f(x_i^-))\}_{i=1}^{k}\right) \right], \tag{3}$$

which encourages alignment of embeddings from the same latent class while pushing them apart from negatives.

**Downstream tasks** The representations learned from the pretraining step can serve as the foundation for downstream supervised tasks such as topic modeling, classification, or sentiment analysis. In unsupervised settings, the extracted representation can be directly utilized for downstream applications, such as anomaly detection or out-of-distribution (OOD) detection (Tran et al., 2022; Vahidi et al., 2024; HaoChen et al., 2021). For simplicity, we'll focus on supervised multi-class classification. The downstream task is defined by $k + 1$ classes represented by the task-specific set $T = \{c_1', \ldots, c_{k+1}'\}$ which is a subset of the full set of downstream task classes, $\mathcal{C}_{\text{down}}$. Examples for task $T$ are generated by the following process :

- sampling a class label $c' \sim \mathcal{D}_T$ (typically uniform),
- drawing an input $x \sim \mathcal{D}_{c'}$.

Given a representation function $f : X \to \mathbb{R}^d$, a linear classifier $W \in \mathbb{R}^{(k+1) \times d}$ is trained on top of the representation function $f$. The supervised loss for a given task $T$ is:

$$\mathcal{L}_{\sup}(T, f) := \inf_{W \in \mathbb{R}^{(k+1) \times d}} \mathcal{L}_{sup}(T, Wf). \tag{4}$$

where

$$\mathcal{L}_{sup}(T, f, W) = \mathbb{E}_{(x,c) \sim \mathcal{D}_T} \left[ \ell \left( \{Wf(x)_c - Wf(x)_{c'}\}_{c' \neq c} \right) \right] \tag{5}$$

and $\ell$ is a margin-based loss function.

To analyze performance more concretely, we consider the *mean classifier*, where the weight vector for each class $c' \in T$ is the mean of its representations $\mu_c := \mathbb{E}_{x \sim \mathcal{D}_c}[f(x)]$:

$$\mathcal{L}_{\sup}^{\mu}(T, f) := \mathcal{L}_{\sup}(T, W^{\mu} f). \tag{6}$$

Here $W^{\mu}$ denotes the *mean classifier*: for each class $c' \in T$, the corresponding row of $W^{\mu}$ is given by the class-mean representation $\mu_c = \mathbb{E}_{x \sim D_c}[f(x)]$. That is, $W^{\mu} \in \mathbb{R}^{(k+1) \times d}$ is defined row-wise as $W_c^{\mu} = \mu_c^{\top}$. This classifier is used only for analysis and provides a deterministic reference model obtained from the representation function $f$.

The average supervised loss over random downstream tasks is:

$$\mathcal{L}^{\mu}_{\sup}(f) := \mathbb{E}_{\{c_i\}_{i=1}^{k+1} \sim \rho^{k+1}} \left[ \mathcal{L}^{\mu}_{\sup}(\{c_i\}, f) \mid c_i \neq c_j \right]. \tag{7}$$

# 3  METHOD AND FRAMEWORK

One of the major challenges in contrastive learning is that the distributions encountered during pretraining rarely match those faced in downstream applications. In practice, models trained on unlabeled data are often evaluated on shifted or perturbed distributions, such as corrupted datasets, new semantic classes, or entirely different domains. This distribution mismatch can degrade the performance of learned representations; yet, most existing theories assume perfect alignment between the pretraining and evaluation distributions. Our objective is to mitigate this gap and provide provable guarantees that explicitly account for distribution shift.

We adopt the latent-class framework of Saunshi et al. (2019) but extend it to account for the misalignment between upstream (pretraining) and downstream distributions.

**Modeling distribution shift and domain generalization.** For *distribution shift*, each pretraining class $c$ may correspond to a downstream class $c'$ whose distribution $\mathcal{D}_{c'}^{\text{down}}$ differs from the pretraining distribution $\mathcal{D}_c$. We define the *mean shift vector*:

$$\delta_c := \mu_c - \mu_c', \quad \mu_c' := \mathbb{E}_{x \sim \mathcal{D}_{c'}^{\text{down}}}[f(x)], \tag{8}$$

with a bounded shift assumption $\|\delta_c\| \leq \epsilon, \forall c$, capturing scenarios where downstream data are drawn from perturbed or shifted versions of pretraining distributions.

Beyond distribution shift, we also consider *domain generalization*, where downstream tasks may involve novel classes not present during pretraining. In this case, $T = \{c_1', \ldots, c_{k+1}'\}$ may include classes outside the pretraining latent class set. For such unseen classes, we model their mean representations directly as $\mu_{c'} := \mathbb{E}_{x \sim \mathcal{D}_{c'}^{\text{down}}}[f(x)], \quad c' \notin \mathcal{C}_{\text{pre}}$, and quantify their discrepancy relative to the pretraining class distribution via the distance between $\mu_{c'}$ and the convex hull of pretraining means $\{\mu_c\}_{c \in \mathcal{C}_{\text{pre}}}$ (See Appendix C).

This unified formulation accounts for both (i) shifts within shared label spaces and (ii) extension to novel label spaces, thereby modeling realistic conditions for out-of-distribution and domain-generalized downstream tasks.

**Contrastive objective.** During pretraining, the encoder $f$ is optimized on unlabeled data drawn from $\mathcal{D}_{\text{sim}}$ and $\mathcal{D}_{\text{neg}}$. The contrastive loss remains

$$\mathcal{L}_{\text{un}}(f) := \mathbb{E}_{(x,x^+) \sim \mathcal{D}_{\text{sim}}, \{x_i^-\} \sim \mathcal{D}_{\text{neg}}^k} \left[ \ell \left( \{f(x)^{\top}(f(x^+) - f(x_i^-))\}_{i=1}^{k} \right) \right]. \tag{9}$$

while the learned representation $\hat{f}$ is obtained by minimizing its empirical estimate.

# 4 GUARANTEED AVERAGE BINARY CLASSIFICATION

## 4.1 UPPER BOUND USING UNSUPERVISED LOSS

We now present our main theoretical result, which establishes a generalization bound for contrastive learning under distribution shift. We first state the result and provide a discussion, followed by the detailed proof. In this section, we prove the guarantee under the assumption of using just 1 negative sample ($k = 1$). We define $L_{sup}(f)$ and $L_{sup}^{\mu}(f)$ as same as Section 2. Before stating and proving the theorem, we define $f_{|S} = (f_t(x_j), f_t(x_j^+), f_t(x_j^-))_{j\in[M],t\in[d]} \in \mathbb{R}^{3dM}$ be the restriction on S for any $f \in \mathcal{F}$. Now, $\mathcal{R}_S(\mathcal{F})$ denotes the Rademacher complexity of the representations $\mathcal{F}$ evaluated on $f_{|S}$.

$$\mathcal{R}_S(\mathcal{F}) = \mathbb{E}_{\sigma\sim\{+1^{3dM}\}}[\sup_{f\in\mathcal{F}} <\sigma, f_{|S}>]$$

Now, let $\tau$ be the probability that independently sampled classes from $\rho$ are same.

$$\tau = \mathbb{E}_{c,c'\sim\rho^2}[1\{c = c'\}]$$

Note that from Section 2 we assumed embeddings are bounded, which means $||f|| \leq R$ for every $f \in \mathcal{F}$ Now we are ready to declare our theorem.

**Theorem 4.1.** *With probability at least $1 - \delta$ over the sampled dataset, the supervised loss under downstream distributions satisfies*

$$L_{sup}^{\mu'}(\hat{f}) \leq \frac{1}{1-\tau}\left(L_{un}(f) - \tau\right) + \frac{Gen_M}{1-\tau} - B(\hat{f}),$$

*where*

$$Gen_M = O\left(\frac{R \cdot \mathfrak{R}_S(\mathcal{F})}{M} + R^2\sqrt{\frac{\log(1/\delta)}{M}}\right),$$

*and the bias term is*

$$B(\hat{f}) = \sup_{\|\delta_c\|\leq\epsilon} \mathbb{E}_{\substack{c^+,c^-\sim\rho^2\\c^+\neq c^-}}\mathbb{E}_{x\sim\mathcal{D}_{c^+}}\left[\ell'\left(f(x)^\top(\mu'_{c^+} - \mu'_{c^-})\right) \cdot f(x)^\top(\delta_{c^+} - \delta_{c^-})\right].$$

**Lemma 4.2** (Generalization of unsupervised loss; (Saunshi et al., 2019)). *With probability at least $1 - \delta$ over the sampled dataset,*

$$L_{un}(\hat{f}) \leq L_{un}(f) + Gen_M,$$

*where*

$$Gen_M = O\left(\frac{R \cdot \mathfrak{R}_S(\mathcal{F})}{M} + R^2\sqrt{\frac{\log(1/\delta)}{M}}\right).$$

*We provide a proof of this lemma in the Appendix A.*

**Lemma 4.3.** *For all $f \in \mathcal{F}$, the supervised loss under downstream (shifted) distributions satisfies:*

$$L_{sup}^{\mu'}(f) \leq \frac{1}{1-\tau}\left(L_{un}(f) - \tau\right) - B(f),$$

*where $L_{sup}^{\mu'}(f)$ is the supervised loss using the mean classifier with downstream distributions $\mu_{c'} = \mathbb{E}_{x\sim\mathcal{D}_{c'}^{down}}[f(x)]$, and $\delta_c := \mu_c - \mu_{c'}$ is the shift vector. While the $B(f)$ is bias term: $B(f) := \mathbb{E}_{\substack{c^+,c^-\sim\rho^2\\c^+\neq c^-}}\mathbb{E}_{x^+\sim\mathcal{D}_{c^+}}\left[\ell'\left(f(x^+)^\top(\mu'_{c^+} - \mu'_{c^-})\right) \cdot f(x^+)^\top(\delta_{c^+} - \delta_{c^-})\right].$*

*Proof.* Our proof extends the original analysis by explicitly incorporating shifted class means and isolating the discrepancy through a first-order Taylor expansion. This extension yields a principled decomposition in which the unsupervised contrastive loss continues to govern downstream performance, augmented by a correction term $B(f)$ that quantifies the effect of distribution shift.

To this end, we start from the definition of the unsupervised contrastive loss:

$$L_{\text{un}}(f) = \mathbb{E}_{(x,x^+)\sim\mathcal{D}_{\text{sim}}, x^-\sim\mathcal{D}_{\text{neg}}}\left[\ell\left(f(x)^\top(f(x^+) - f(x^-))\right)\right].$$

As in the original proof, we reparameterize this expectation in terms of class-level sampling:

$$L_{\text{un}}(f) = \mathbb{E}_{c^+,c^-\sim\rho^2}\mathbb{E}_{x^+\sim\mathcal{D}_{c^+}, x^-\sim\mathcal{D}_{c^-}}\left[\ell\left(f(x)^\top(f(x^+) - f(x^-))\right)\right].$$

Now, we apply Jensen's inequality, accounting for the shifted distributions used by the downstream supervised task. Instead of approximating with $\mu_c$, we use $\mu_{c'}$ and explicitly account for the shift via $\delta$. This yields:

$$\geq \mathbb{E}_{c^+,c^-\sim\rho^2}\mathbb{E}_{x\sim\mathcal{D}_{c^+}}\left[\ell\left(f(x)^\top(\mu'_{c^+} + \delta_{c^+} - \mu'_{c^-} - \delta_{c^-})\right)\right].$$

We now split the expectation into the two cases:

$$= \tau\cdot\mathbb{E}_{c\sim\rho}\mathbb{E}_{x\sim\mathcal{D}_c}\left[\ell\left(f(x)^\top(0)\right)\right] + (1-\tau)\cdot\mathbb{E}_{\substack{c^+,c^-\sim\rho^2 \\ c^+\neq c^-}}\mathbb{E}_{x\sim\mathcal{D}_{c^+}}\left[\ell\left(f(x)^\top(\mu'_{c^+} - \mu'_{c^-} + \delta_{c^+} - \delta_{c^-})\right)\right].$$

Letting $w = f(x)^\top(\mu'_{c^+} - \mu'_{c^-})$ and $\Delta = f(x)^\top(\delta_{c^+} - \delta_{c^-})$, we apply a first-order Taylor expansion:

$$\ell(w + \Delta) \approx \ell(w) + \ell'(w)\cdot\Delta.$$

Thus:

$$L_{\text{un}}(f) \geq \tau\cdot\ell(0) + (1-\tau)\cdot\mathbb{E}_{\substack{c^+,c^-\sim\rho^2 \\ c^+\neq c^-}}\mathbb{E}_{x\sim\mathcal{D}_{c^+}}\left[\ell\left(f(x)^\top(\mu'_{c^+} - \mu'_{c^-})\right) + \ell'\left(f(x)^\top(\mu'_{c^+} - \mu'_{c^-})\right)\cdot f(x)^\top(\delta_{c^+} - \delta_{c^-})\right].$$

We define:

$$L_{\text{sup}}^{\mu'}(f) := \mathbb{E}_{\substack{c^+,c^-\sim\rho^2 \\ c^+\neq c^-}}\mathbb{E}_{x\sim\mathcal{D}_{c^+}}\left[\ell\left(f(x)^\top(\mu'_{c^+} - \mu'_{c^-})\right)\right],$$

$$B(f) := \sup_{\substack{\delta_{c^+},\delta_{c^-} \\ \|\delta_c\|\leq\epsilon}} \mathbb{E}_{\substack{c^+,c^-\sim\rho^2 \\ c^+\neq c^-}}\mathbb{E}_{x\sim\mathcal{D}_{c^+}}\left[\ell'\left(f(x)^\top(\mu'_{c^+} - \mu'_{c^-})\right)\cdot f(x)^\top(\delta_{c^+} - \delta_{c^-})\right]$$

So we have:

$$L_{\text{un}}(f) \geq (1-\tau)\cdot\left(L_{\text{sup}}^{\mu'}(f) + B(f)\right) + \tau.$$

Rearranging gives the result:

$$L_{\text{sup}}^{\mu'}(f) \leq \frac{1}{1-\tau}\left(L_{\text{un}}(f) - \tau\right) - B(f). \quad\blacksquare$$

$$\square$$

The result follows directly by applying Lemma 4.3 for $\hat{f}$ and finishing up with Lemma 4.2.

Now as mentioned by (Saunshi et al., 2019), One could argue that if $\mathcal{F}$ is rich enough such that $L_u n$ can be made small, then Theorem 4.1 suffices. However, in the next section we explain that unless $\tau \ll 1$, this may not always be possible and we show one way to alleviate this.

## 4.2 PRICE OF NEGATIVE SAMPLING: CLASS COLLISION

It's important to mention which the unsupervised loss can be broken down into its components as follows:

$$L_{un}(f) = \tau L_{un}^=(f) + (1-\tau)L_{un}^{\neq}(f) \tag{10}$$

where $L_{un}^{\neq}(f)$ represents the loss suffered when the similar pair and the negative example come from different classes.

$$L_{un}^{\neq}(f) = \mathbb{E}_{\substack{c^+,c^-\sim\rho^2 \\ x,x^+\sim\mathcal{D}_{c^+} \\ x^-\sim\mathcal{D}_{c^-}}}\left[l(f(x)^T(f(x^+) - f(x^-)))|c^+ \neq c^-\right]$$

while $L_{un}^=(f)$ is when they come from the same class.

Consider $\nu$ as a distribution over $\mathcal{C}$ with $\nu(c) \propto \rho^2(c)$, then

$$L_{un}^{=}(f) = \mathbb{E}_{\substack{c \sim \nu \\ x,x^+,x^- \sim \mathcal{D}_c^3}}[l(f(x)^T(f(x^+) - f(x^-)))] \geq \mathbb{E}_{c \sim \nu, x \sim \mathcal{D}_c}[l(f(x)^T(\mu_c - \mu_c))] = 1$$

by Jensen's inequality again, which implies $L_{un}(f) \geq \tau$. In general, without any further assumptions on $f$, $L_{un}(f)$ can be far from $\tau$, rendering the bound in Theorem 4.1 useless. However, as we will show, the magnitude of $L_{un}^{=}(f)$ can be controlled by the intra-class deviation of $f$. Let $\Sigma(f,c)$ the covariance matrix of $f(x)$ when $x \sim \mathcal{D}_c$. We define a notion of intra-class deviation as:

$$s(f) := \mathbb{E}_{c \sim \nu}[\sqrt{||\Sigma(f,c)||_2}\mathbb{E}_{x \sim \mathcal{D}_c}||f(x)||]$$

**Lemma 4.4.** *For all $f \in \mathcal{F}$*

$$L_{un}^{=}(f) - 1 \leq c's(f)$$

*where $c'$ is a positive constant.*

We prove Lemma 4.4 in Appendix A. Theorem 4.1 combined with Equation 10 and Lemma 4.4 gives the following result :

**Theorem 4.5.** *With probability at least $1 - \delta$, $\forall f \in \mathcal{F}$*

$$L_{sup}(\hat{f}) \leq L_{sup}^{\mu}(\hat{f}) \leq L_{un}^{\neq}(f) + \beta s(f) + \eta Gen_M - B(\hat{f}) \tag{11}$$

*where $\beta = c'\frac{\tau}{1-\tau}$, $\eta = \frac{1}{1-\tau}$ and $c'$ is a constant and $B(\hat{f})$ is our bias term.*

The above bound highlights two sufficient properties of the function class for unsupervised learning to work: when the function class $\mathcal{F}$ is rich enough to contain some $f$ with low $\beta s(f)$ as well as low $L_{un}^{\neq}(f)$ then $\hat{f}$, the empirical minimizer of the unsupervised loss—learned using sufficiently large number of samples—will have good performance on supervised tasks (low $L_{sup}(\hat{f})$).

**Discussion.** The question raised: how does a distribution shift affect supervised downstream loss? The following result extends the mean-classifier analysis of (Saunshi et al., 2019)) by explicitly incorporating the discrepancy between latent pretraining distributions and downstream task distributions. The bound reveals that the supervised error of the mean classifier is controlled not only by the unsupervised contrastive loss $L_{un}(f)$ but also by an additional term $B(f)$ that quantifies the impact of distribution shift.

In particular, when downstream classes are perfectly aligned with the latent classes (i.e., $\delta_c = 0$ for all $c$), the bias term vanishes and our bound reduces to the original in-distribution guarantee.

More generally, when $\|\delta_c\| \leq \epsilon$, the bias can be bounded in terms of $\epsilon$, the Lipschitz constant $L$ of the loss, and either the norm bound $R$ on representations or the sub-Gaussian width $\sigma_f$. This shows that the effect of distribution shift grows at most linearly with $\epsilon$, and is further moderated by the regularity of the encoder distribution. Intuitively, the closer the downstream class means $\mu_{c'}$ are to the latent class means $\mu_c$, the smaller the penalty from shift and the more reliable the transferred representations. Conversely, large shifts directly degrade the guarantee, reflecting the difficulty of domain generalization.

## 5 GUARANTEES FOR k NEGATIVE SAMPLES

In this section we explore two extensions to our analysis. First, in Section 5.1, inspired by empirical works like Logeswaran & Lee (2018) that often use more than one negative sample for every similar pair, we show provable guarantees for this case by careful handling of class collision.

### 5.1 GUARANTEES FOR $k$ NEGATIVE SAMPLES

Here, the algorithm employs $k$ negative samples $x_1^-, \ldots, x_k^-$ drawn i.i.d. from $D_{neg}$ for every positive sample pair $(x, x^+) \sim D_{sim}$ and minimizes Equation (3). As in Section 4, we prove a bound for $\hat{f}$ of the following form:

**Theorem 5.1** (Informal version). *For all $f \in \mathcal{F}$,*

$$L_{\sup}(\hat{f}) \leq L_{\sup}^{\mu}(\hat{f}) \leq \alpha L_{un}^{\neq}(f) + \beta s(f) + \eta \left(Gen_M - B(f)\right),$$

*where $L_{un}^{\neq}(f)$ and $Gen_M$ and $B(f)$ are extensions of the corresponding terms from Section 4, and $s(f)$ remains unchanged.*

The formal statement of the theorem and its proof appears in Appendix B. The key differences from Theorem 4.5 are the coefficient $\beta$ and the distribution of tasks in $L_{\sup}$ that we describe below. The coefficient $\beta$ of $s(f)$ increases with $k$, e.g. when $\rho$ is uniform and $k \ll |\mathcal{C}|$,

$$\beta \approx \frac{k}{|\mathcal{C}|}.$$

The average supervised loss that we bound is

$$L_{\sup}(\hat{f}) := \mathbb{E}_{T \sim \mathcal{D}}\left[L_{\sup}(T, \hat{f})\right],$$

where $\mathcal{D}$ is a distribution over tasks, defined as follows: sample $k+1$ classes $c^+, c_1^-, \ldots, c_k^- \sim \rho^{k+1}$, conditioned on the event that $c^+$ does not also appear as a negative sample. Then, set $T$ to be the set of distinct classes in $\{c^+, c_1^-, \ldots, c_k^-\}$. $L_{\sup}^{\mu}(\hat{f})$ is defined by using $L_{\sup}^{\mu}(T, \hat{f})$.

**Remark.** Bounding $L_{\sup}(\hat{f})$ directly gives a bound for average $(k+1)$-way classification loss $L_{\sup}(\hat{f})$ from Definition 2.2, since

$$L_{\sup}(\hat{f}) \leq \frac{L_{\sup}(\hat{f})}{p},$$

where $p$ is the probability that the $k+1$ sampled classes are distinct. For $k \ll |\mathcal{C}|$ and $\rho \approx$ uniform, these metrics are almost equal.

## 6    Theoretical Analysis

The key quantity $B(f)$ determines how robust contrastive learning is under shift. We bound it under various assumptions on the loss function $\ell$. We assume throughout that the feature encoder satisfies $\|f(x)\| \leq R$ for all $x \in \mathcal{X}$, and that the shift vector between latent and downstream class means is bounded: $\|\delta_c\| \leq \epsilon$.

**General $L$-Lipschitz Loss.**    Assume the loss function $\ell$ is $L$-Lipschitz. Then for any $z, \delta \in \mathbb{R}$, we have:

$$|\ell(z + \delta) - \ell(z)| \leq L \cdot |\delta|.$$

In our case, the shift introduces a bias $\Delta = f(x)^\top (\delta_{c^+} - \delta_{c^-})$, so we bound:

$$|f(x)^\top (\delta_{c^+} - \delta_{c^-})| \leq \|f(x)\| \cdot \|\delta_{c^+} - \delta_{c^-}\| \leq 2R\epsilon.$$

Thus, the loss deviation is bounded by:

$$B(f) \leq L \cdot 2R\epsilon.$$

**Hinge Loss.**    For hinge loss $\ell(z) = \max(0, 1 - z)$, the subgradient satisfies:

$$\ell'(z) \in \begin{cases} [-1, 0] & \text{if } z = 1, \\ -1 & \text{if } z < 1, \\ 0 & \text{if } z > 1. \end{cases}$$

So $|\ell'(z)| \leq 1$. Therefore:

$$B(f) \leq \sup_{\|\delta_c\| \leq \epsilon} \mathbb{E}_{c^+, c^-} \mathbb{E}_{x \sim \mathcal{D}_{c^+}} \left[|f(x)^\top (\delta_{c^+} - \delta_{c^-})|\right] \leq 2R\epsilon.$$

This gives the bound:

$$L_{\sup}^{\mu'}(f) \leq \frac{1}{1 - \tau} \left(L_{un}(f) - \tau\right) - 2R\epsilon.$$

**Logistic Loss.** For the logistic loss $\ell(z) = \log(1 + e^{-z})$, we have:

$$\ell'(z) = -\frac{1}{1 + e^z} \in (-1, 0),$$

so it is 1-Lipschitz and we again have $|\ell'(z)| \leq 1$. Thus, the bound is the same as the hinge loss case:

$$B(f) \leq 2R\epsilon,$$

and the full inequality becomes:

$$L_{\mathrm{sup}}^{\mu'}(f) \leq \frac{1}{1 - \tau} \left( L_{\mathrm{un}}(f) - \tau \right) - 2R\epsilon.$$

**Bound under Lipschitz Loss and Sub-Gaussian Representation.** Assume the loss function $\ell$ is $L$-Lipschitz, and that the representation function $f(x) \in \mathbb{R}^d$ is sub-Gaussian with parameter $\sigma_f$, meaning for all unit vectors $u \in \mathbb{R}^d$, the projection $u^\top f(x)$ satisfies:

$$\mathbb{P}(|u^\top f(x)| \geq t) \leq 2 \exp\left(-\frac{t^2}{2\sigma_f^2}\right).$$

Then the distribution shift bias satisfies:

$$B(f) \leq 2\epsilon \cdot L \cdot \sigma_f \sqrt{2/\pi},$$

and the overall supervised loss satisfies:

$$L_{\mathrm{sup}}^{\mu'}(f) \leq \frac{1}{1 - \tau} \left( L_{\mathrm{un}}(f) - \tau \right) - 2\epsilon \cdot L \cdot \sigma_f \sqrt{2/\pi}.$$

These results show that the penalty induced by shift scales linearly with both the encoder norm and the shift magnitude $\epsilon$, but can be significantly tighter when $f(x)$ concentrates in high dimensions.

**Estimating the shift in practice.** Since pretraining is unsupervised, $\mu_c$ is not directly observable, making $\delta_c$ difficult to estimate. However, approximate strategies exist: (i) averaging positive pairs (via augmentations) provides a Monte Carlo estimate of latent means, (ii) clustering embeddings recovers pseudo-classes, and (iii) large pretrained models often induce strong semantic grouping that facilitates alignment. While exact estimation is impossible without labels, such heuristics can produce meaningful approximations of $\delta_c$, enabling tighter, data-dependent bounds in practice.

**Summary.** Under all these assumptions, the supervised loss under shifted downstream distributions can be upper bounded by:

$$L_{\mathrm{sup}}^{\mu'}(f) \leq \frac{1}{1 - \tau} \left( \widehat{L}_{\mathrm{un}}(f) + \beta_M - \tau \right) - B(f),$$

where $\beta_M$ accounts for generalization from the empirical to the population contrastive loss. This inequality quantifies the robustness of contrastive learning to bounded distribution shift and highlights the role of encoder regularization in mitigating the shift bias.

## 7 EMPIRICAL ANALYSIS

Our theoretical results show the bias term $B(f)$ as a central quantity governing OOD generalization. Specifically, $B(f)$ relies on the deviation between pretraining class means and downstream class means. To empirically validate this relationship, we approximate the associated mean-shift term and explore how it correlates with downstream performance across both synthetic and real-world shifts. We consider three experimental settings: (i) when the pretraining distribution is known and accessible, (ii) when pretraining data are unavailable and class structure must be inferred from unlabeled embeddings, and (iii) a real-world domain generalization setting on the PACS dataset to verify our bounds under natural distribution shifts (detailed in Appendix D). These experiments collectively evaluate whether the geometric shift predicted by $B(f)$ indeed governs transfer accuracy. In section 4.1, we observed that the bias term takes the form

$$B(\hat{f}) \;=\; \sup_{\substack{\|\delta_c\| \leq \epsilon}} \mathbb{E}_{\substack{c^+, c^- \sim \rho^2 \\ c^+ \neq c^-}} \mathbb{E}_{x \sim \mathcal{D}_{c^+}} \left[ \ell'\big(f(x)^\top(\mu'_{c^+} - \mu'_{c^-})\big) \cdot f(x)^\top(\delta_{c^+} - \delta_{c^-}) \right]. \tag{12}$$

The inner product term

$$f(x)^\top(\delta_{c^+} - \delta_{c^-})$$

quantifies how differences in the class means directly influence the $B(f)$ and our generalization bound. Therefore, the shift vector $\delta_c$ plays a central role in determining the transferability of $f$.

To empirically probe this effect, our experiments estimate the shift vector as introduced in section 4.1 by measuring the displacement between class means in the pretraining distribution and the downstream distribution:

$$\|\delta_c\| \;\approx\; \big\|\mu_c^{\mathrm{pre}} - \mu_c^{\mathrm{down}}\big\|_2,$$

so that the empirical mean-shift quantity we measure is

$$\widehat{\Delta}(f) \;=\; \frac{1}{C}\sum_{c=1}^{C}\big\|\widehat{\mu}_c^{\mathrm{pre}} - \widehat{\mu}_c^{\mathrm{down}}\big\|_2, \tag{13}$$

which provides a data-driven proxy for the size of the perturbations $\delta_c$ appearing in $B(\hat{f})$ (Eq. 12). Thus, by examining the relationship between $\widehat{\Delta}(f)$ and downstream accuracy, our experiments directly evaluate how the structure of $B(\hat{f})$ predicts transfer performance.

**Mean classifier.** As defined in Section 2, the downstream linear classifier $W^\mu$ assigns to each class $c$ the weight vector

$$W_c^\mu = \mu_c^{\mathrm{down}} = \mathbb{E}_{x \sim \mathcal{D}_c^{\mathrm{down}}} f(x),$$

and predicts with

$$W^\mu(x) = \arg\max_{c \in [C]} \langle f(x), W_c^\mu \rangle.$$

Since $W_c^\mu$ depends directly on downstream class means, this classifier is the natural empirical proxy for evaluating the effect of mean perturbations inside $B(\hat{f})$.

## 7.1 Scenario 1: Known Pretraining Distribution (Oracle Access)

In the first experiment, we assume that the encoder $f$ was contrastively pretrained on CIFAR-10 and we *have access to the pretraining samples themselves*. This assumption may not hold in real-world foundation-model settings, but it provides a controlled environment for isolating the geometric effect of distribution shift.

To this end, we consider the self-supervised encoder of ResNet-34 trained contrastively on CIFAR-10 and compute the empirical pretraining class means $\widehat{\mu}_c^{\mathrm{pre}}$. On the other hand, for each of the 15 corruption types in CIFAR-10-C and each severity level $s \in \{1, \ldots, 5\}$, we embed the corrupted images and compute the downstream means $\widehat{\mu}_c^{\mathrm{down}}(s)$. We compute the mean shift $\widehat{\Delta}(f, s)$ and evaluate the oracle mean classifier $W^{\mu^{\mathrm{pre}}}$ on the downstream test split.

As illustrated in Fig. 1a, across all corruptions and severities, we observe a clear and approximately linear *negative correlation* between mean shift and accuracy. Higher severities induce larger feature-space displacements between $\widehat{\mu}_c^{\mathrm{pre}}$ and $\widehat{\mu}_c^{\mathrm{down}}(s)$, and the performance of $W^{\mu^{\mathrm{pre}}}$ drops proportionally. Hence, $\Delta(f)$ is one of the principal contributors to the bias term $B(f)$. These results provide direct empirical evidence that $B(f)$ correctly captures the transfer difficulty predicted by our theory.

## 7.2 Scenario 2: Unknown Pretraining Distribution (Unsupervised Mean Recovery)

The second experiment reflects the realistic situation in which the encoder $f$ is obtained through self-supervised pretraining on an unknown and unavailable dataset. In this case, neither $D^{\mathrm{pre}}$ nor its class means are accessible. We therefore estimate class structure by clustering.

Here, the encoder is a SimCLR Chen et al. (2020) trained in a fully contrastive manner without labels. We do not assume that CIFAR-10 was part of its pretraining data. We embed CIFAR-10

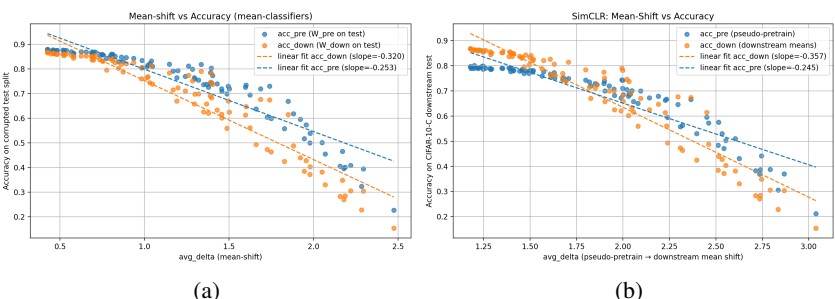

|  (a)  |  (b)  |

Figure 1: Downstream accuracy of CIFAR-10-C versus mean shift for the two experimental scenarios. **(a)** With oracle access to CIFAR-10 pretraining means (contrastively trained ResNet-34), downstream accuracy on CIFAR-10-C decreases smoothly as the mean shift grows, reflecting the effect predicted by the bias term $B(\hat{f})$. **(b)** When the pretraining distribution is unknown, pseudo-class means recovered via $K$-means exhibit the same relationship: As shift vector $\delta$ grows, downstream accuracy drops. This validates the predictive role of mean-shift in $B(\hat{f})$ under both supervised and unsupervised scenarios.

(without using its labels) and perform $K$-means with $K = 10$ to obtain pseudo-classes. For each cluster $k$, we compute a pseudo-mean $\widehat{\mu}_k^{\text{pre}} = \frac{1}{|\mathcal{C}_k|} \sum_{x \in \mathcal{C}_k} f(x)$. We apply the Hungarian algorithm to find the optimal cluster-to-class matching to align cluster indices with downstream classes. For each corruption type and severity, we compute the downstream means and the corresponding mean shift $\widehat{\Delta}_{\text{unsup}}(f, s)$. We evaluate the pseudo-mean classifier $W^{\hat{\mu}}$ on the downstream test dataset.

Even though pretraining labels and pretraining data are unavailable, the clustering-based pseudo-means recover sufficient structure to reproduce the same qualitative relationship observed in Scenario 1: *larger mean shifts lead to lower downstream accuracy*. Thus, the dominant component of $B(f)$—the geometric mismatch between pretraining and downstream class means—remains predictive even when estimated via unsupervised pseudo-labels.

These empirical results (Fig. 1) support our theoretical claim that $B(f)$ captures the intrinsic transfer difficulty of a representation, independent of how or where the encoder was trained.

Both experimental scenarios reveal a consistent phenomenon: (i) Mean shift, the key quantity inside $B(f)$, tightly predicts downstream accuracy across corruption types and severities. (ii) When pretraining data are known, the measured $\widehat{\Delta}(f)$ precisely tracks the degradation predicted by our bounds. (iii) When pretraining data are unknown, a clustering-based approximation of class means still yields the same monotonic relationship, demonstrating robustness of the theory.

While our primary analysis focuses on controlled shifts using CIFAR-10-C, we further validate our framework on the PACS benchmark, a standard dataset for real-world domain generalization. In Appendix D, we demonstrate that our theoretical bounds and the shift magnitude $\delta$ accurately predict performance drops across diverse domains (Photo, Art Painting, Cartoon, Sketch), confirming that our findings extend beyond synthetic corruptions.

## 8 CONCLUSION AND FUTURE WORKS

In this paper, we introduced a theoretical framework that extends the latent class model of contrastive learning to incorporate both distributional shift and domain generalization, two central challenges in real-world transfer settings. Our analysis reveals specific bias and variance terms in the downstream supervised loss, enabling us to clearly characterize how representational misalignment affects generalization. We also demonstrated that the bias can be bounded under realistic conditions related to the encoder and the loss function, which in turn provides provable guarantees for a wide range of contrastive methods. Beyond advancing theoretical understanding, our framework unifies prior analyses of contrastive learning with more general transfer scenarios, covering settings where downstream tasks draw from shifted or expanded label spaces. We expect this approach to influence the design of more robust pretraining objectives and inform empirical evaluations of representation transferability. Future research may explore extensions to other self-supervised paradigms, as well as investigating tighter relations between theoretical bounds and practical performance.

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

## A GENERALIZATION BOUND

We first state the following general Lemma in order to bound the generalization error of the function class $\mathcal{F}$ on the unsupervised loss function $L_{un}(\cdot)$ lemma 4.2 can be directly derived from it.

**Lemma A.1.** *Let $l : \mathbb{R}^k \to \mathbb{R}$ be $\eta$-Lipschitz and bounded by $B$. Then with probability at least $1 - \delta$ over the training set $\mathbb{S} = \{(x_j, x_j^+, x_{j1}^-, \ldots, x_{jk}^-)\}_{j=1}^M$, for all $f \in \mathcal{F}$*

$$L_{un}(\hat{f}) \le L_{un}(f) + O\left(\eta R \sqrt{k} \frac{\mathcal{R}_{\mathcal{S}}(\mathcal{F})}{M} + B\sqrt{\frac{\log\frac{1}{\delta}}{M}}\right) \tag{14}$$

*where*

$$\mathcal{R}_{\mathcal{S}}(\mathcal{F}) = \mathbb{E}_{\sigma \sim \{\pm 1\}^{(k+2)dM}}\left[\sup_{f \in \mathcal{F}} \langle \sigma, f_{|\mathbb{S}} \rangle \right] \tag{15}$$

*and $f_{|\mathbb{S}} = (f_t(x_j), f_t(x_j^+), f_t(x_{j1}^-), \ldots, f_t(x_{jk}^-))_{j \in [M], t \in [d]}$.*

Note that for $k + 1$-way classification, for hinge loss we have $\eta = 1$ and $B = O(R^2)$, while for logistic loss $\eta = 1$ and $B = O(R^2 + \log k)$. Setting $k = 1$, we get Lemma **??**. We now prove A.1.

*Proof of A.1.* First, we use the classical bound for the generalization error in terms of the Rademacher complexity (see (Mohri et al., 2018) Theorem 3.1). For a real function class $\mathcal{G}$ whose functions map from a set $\mathcal{Z}$ to $[0, 1]$ and for any $\delta > 0$, if $\mathbb{S}$ is a training set composed by $M$ iid samples $\{z_j\}_{j=1}^M$, then with probability at least $1 - \frac{\delta}{2}$, for all $g \in \mathcal{G}$

$$\mathbb{E}[g(z)] \le \frac{1}{M} \sum_{j=1}^M g(z_i) + 2\frac{\mathcal{R}_{\mathcal{S}}(\mathcal{G})}{M} + 3\sqrt{\frac{\log\frac{4}{\delta}}{2M}} \tag{16}$$

where $\mathcal{R}_{\mathcal{S}}(\mathcal{G})$ is the usual Rademacher complexity. We apply this bound to our case by setting $\mathcal{Z} = \mathcal{X}^{k+2}$, $\mathbb{S}$ is our training set and the function class is

$$\mathcal{G} = \left\{ g_f(x, x^+, x_1^-, \ldots, x_k^-) = \frac{1}{B} \ell\left(\{f(x)^T(f(x^+) - f(x_i^-))\}_{i=1}^k\right) \,\middle|\, f \in \mathcal{F} \right\} \tag{17}$$

We will show that for some universal constant c, $\mathcal{R}_{\mathcal{S}}(\mathcal{G}) \le \frac{c\eta R \sqrt{k}}{B}\mathcal{R}_{\mathcal{S}}(\mathcal{F})$ or equivalently

$$\mathbb{E}_{\sigma \sim \{\pm 1\}^M}\left[\sup_{f \in \mathcal{F}} \langle \sigma, (g_f)_{|\mathbb{S}} \rangle\right] \le \frac{c\eta R \sqrt{k}}{B}\mathbb{E}_{\sigma \sim \{\pm 1\}^{d(k+2)M}}\left[\sup_{f \in \mathcal{F}} \langle \sigma, f_{|\mathbb{S}} \rangle\right] \tag{18}$$

where $(g_f)_{|\mathbb{S}} = \{g_f(x_j, x_j^+, x_{j1}^-, ..., x_{jk}^-)\}_{j=1}^M$. To do that we will use the following vector-contraction inequality.

**Theorem A.2** ((Corollary 4 in (Maurer, 2016))). *Let $\mathcal{Z}$ be any set, and $\mathbb{S} = \{z_j\}_{j=1}^M \in \mathcal{Z}^M$. Let $\tilde{\mathcal{F}}$ be a class of functions $\tilde{f} : \mathcal{Z} \to \mathbb{R}^n$ and $h : \mathbb{R}^n \to \mathbb{R}$ be $L$-Lipschitz. For all $\tilde{f} \in \tilde{\mathcal{F}}$, let $g_{\tilde{f}} = h \circ \tilde{f}$. Then*

$$\mathbb{E}_{\sigma \sim \{\pm 1\}^M}\left[\sup_{\tilde{f} \in \tilde{\mathcal{F}}} \langle \sigma, (g_{\tilde{f}})_{|\mathbb{S}} \rangle\right] \le \sqrt{2}L\mathbb{E}_{\sigma \sim \{\pm 1\}^{nM}}\left[\sup_{\tilde{f} \in \tilde{\mathcal{F}}} \langle \sigma, \tilde{f}_{|\mathbb{S}} \rangle\right]$$

*where $\tilde{f}_{|\mathbb{S}} = (\tilde{f}_t(z_j))_{t \in [n], j \in [M]}$.*

We apply Theorem A.2 to our case by setting $\mathcal{Z} = \mathcal{X}^{k+2}$, $n = d(k + 2)$ and

$$\tilde{\mathcal{F}} = \{\tilde{f}(x, x^+, x_{j1}^-, \ldots, x_{jk}^-) = (f(x), f(x^+), f(x_{j1}^-), \ldots, f(x_{jk}^-)) | f \in \mathcal{F}\}$$

We also use $g_{\tilde{f}} = g_f$ where $\tilde{f}$ is derived from $f$ in the definition of $\tilde{F}$. Observe that now A.2 is exactly in the form of 18 and we need to show that $L \le \frac{c\sqrt{2}\eta R \sqrt{k}}{B}$ for some constant c. But,

for $z = (x, x^+, x_1^-, \ldots, x_k^-)$, we have $g_{\tilde{f}}(z) = \frac{1}{B}\ell(\phi(\tilde{f}(z)))$ where $\phi : \mathbb{R}^{(k+2)d} \to \mathbb{R}^k$ and $\phi((v_t, v_t^+, v_{t1}^-, \ldots, v_{tk}^-)_{t \in [d]}) = (\sum_t v_t(v_t^+ - v_{ti}^-))_{i \in [k]}$. Thus, we may use $h = \frac{1}{B}l \circ \phi$ to apply Theorem A.2.

Now, we see that $\phi$ is $\sqrt{6k}R$-Lipschitz when $\sum_t v_t^2, \sum_t (v_t^+)^2, \sum_t (v_{tj}^-)^2 \leq R^2$ by computing its Jacobian. Indeed, for all $i, j \in [k]$ and $t \in [d]$, we have $\frac{\partial \phi_i}{\partial v_t} = v_t^+ - v_{ti}^-, \frac{\partial \phi_i}{\partial v_t^+} = v_t$ and $\frac{\partial \phi_i}{\partial v_{tj}^-} = -v_t \mathbb{1}\{i = j\}$. From the triangle inequality, the Frobenius norm of the Jacobian $J$ of $\phi$ is

$$||J||_F = \sqrt{\sum_{i,t}(v_t^+ - v_{ti}^-)^2 + 2k\sum_t v_t^2} \leq \sqrt{4kR^2 + 2kR^2} = \sqrt{6k}R$$

Now, taking into account that $||J||_2 \leq ||J||_F$, we have that $\phi$ is $\sqrt{6k}R$-Lipschitz on its domain and since $l$ is $\eta$-Lipschitz, we have $L \leq \frac{\sqrt{6}\eta R \sqrt{k}}{B}$.

Now, we have that with probability at least $1 - \frac{\delta}{2}$

$$L_{un}(\hat{f}) \leq \hat{L}_{un}(\hat{f}) + O\left(\frac{\eta R \sqrt{k}\mathcal{R}_S(\mathcal{F})}{M} + B\sqrt{\frac{\log\frac{1}{\delta}}{M}}\right) \tag{19}$$

Let $f^* \in \arg\min_{f \in \mathcal{F}} L_{un}(f)$. With probability at least $1 - \frac{\delta}{2}$, we have that $\hat{L}_{un}(f^*) \leq L_{un}(f^*) + 3B\sqrt{\frac{\log\frac{2}{\delta}}{2M}}$ (Hoeffding's inequality). Combining this with Equation 19, the fact that $\hat{L}_{un}(\hat{f}) \leq \hat{L}_{un}(f^*)$ and applying a union bound, finishes the proof. $\qquad\square$

## A.2. CLASS COLLISION LEMMA

We prove a general Lemma, from which Lemma 4.4 can be derived directly.

**Lemma A.3.** *Let $c \in \mathcal{C}$ and $\ell : \mathbb{R}^t \to \mathbb{R}$ be either the t-way hinge loss or t-way logistic loss. Let $x, x^+, x_1^-, \ldots, x_t^-$ be iid draws from $\mathcal{D}_c$. For all $f \in \mathcal{F}$, let*

$$L_{un,c}^=(f) = \mathbb{E}_{x,x^+,x_i^-}\left[\ell\left(\{f(x)^T(f(x^+) - f(x_i^-))\}_{i=1}^t\right)\right]$$

*Then*

$$L_{un,c}^=(f) - \ell(\vec{0}) \leq c_0 t \sqrt{||\Sigma(f,c)||_2}\mathbb{E}_{x \sim \mathcal{D}_c}[||f(x)||] \tag{20}$$

*where $c_0$ is a positive constant.*

Lemma 4.4 is a direct consequence of the above Lemma, by setting $t = 1$ (which makes $\ell(0) = 1$), taking an expectation over $c \sim \nu$ in Equation 20 and noting that $\mathbb{E}_{c \sim \nu}[L_{un,c}^=(f)] = L_{un}^=(f)$.

*Proof of Lemma A.3.* Fix an $f \in \mathcal{F}$ and let $z_i = f(x)^T(f(x_i^-) - f(x^+))$ and $z = \max_{i \in [t]} z_i$. First, we show that

$$L_{un,c}^=(f) - \ell(\vec{0}) \leq c_0 \mathbb{E}[|z|], \text{ for some constant } c_0.$$

Note that $\mathbb{E}[|z|] = P[z \geq 0]\mathbb{E}[z|z \geq 0] + P[z \leq 0]\mathbb{E}[-z|z \leq 0] \geq P[z \geq 0]\mathbb{E}[z|z \geq 0]$.

**t-way hinge loss:** By definition $\ell(v) = \max\{0, 1 + \max_{i \in [t]}\{-v_i\}\}$. Here, $L_{un,c}^=(f) = \mathbb{E}[(1 + z)_+] \leq \mathbb{E}[\max\{1 + z, 1\}] = 1 + P[z \geq 0]\mathbb{E}[z|z \geq 0] \leq 1 + \mathbb{E}[|z|]$.

**t-way logistic loss:** By definition $\ell(v) = \log_2(1 + \sum_{i=1}^t e^{-v_i})$, we have $L_{un,c}^=(f) = \mathbb{E}[\log_2(1 + \sum_i e^{z_i})] \leq \mathbb{E}[\log_2(1 + te^z)] \leq \max\{\frac{z}{\log 2} + \log_2(1 + t), \log_2(1 + t)\} = \frac{P[z \geq 0]\mathbb{E}[z|z \geq 0]}{\log 2} + \log_2(1 + t) \leq \frac{\mathbb{E}[|z|]}{\log 2} + \log_2(1 + t)$.

Finally, $\mathbb{E}[|z|] \leq \mathbb{E}[\max_{i \in [t]}|z_i|] \leq t\mathbb{E}[|z_1|]$. But,

$$\mathbb{E}[|z_1|] = \mathbb{E}_{x,x^+,x_1^-}\left[|f(x)^T(f(x_1^-) - f(x^+))|\right]$$

$$\leq \mathbb{E}_x \left[ \|f(x)\| \sqrt{\mathbb{E}_{x^+,x_1^-} \left[ \left( \frac{f(x)^T}{\|f(x)\|} (f(x_1^-) - f(x^+)) \right)^2 \right]} \right] \leq \sqrt{2}\sqrt{\|\Sigma(f,c)\|_2} \mathbb{E}_{x \sim \mathcal{D}_c}[\|f(x)\|]$$

$\square$

## B  MODIFIED PROOF UNDER DISTRIBUTIONAL SHIFT

We now present Theorem B.1 as the formal statement of Theorem 5.1 and prove it. First, we define some necessary quantities. Let $(c^+, c_1^-, \ldots, c_k^-)$ be $k+1$ not necessarily distinct classes. We define $Q(c^+, c_1^-, \ldots, c_k^-)$ to be the set of distinct classes in this tuple. We also define $I^+(c_1^-, \ldots, c_k^-) = \{i \in [k] | c_i^- = c^+\}$ to be the set of indices where $c^+$ reappears in the negative samples . We will abuse notation and just write $Q, I^+$ when the tuple is clear from the context.

To define $L_{un}^{\neq}(f)$ consider the following tweak in the way the latent classes are sampled: sample $c^+, c_1^-, \ldots, c_k^- \sim \rho^{k+1}$ conditioning on $|I^+| < k$ and then remove all $c_i^-, i \in I^+$ . The datapoints are then sampled as usual: $x, x^+ \sim \mathcal{D}_{c^+}^2$ and $x_i^- \sim \mathcal{D}_{c_i^-}, i \in [k]$, independently.

$$L_{un}^{\neq}(f) := \mathbb{E}_{c^+, c_i^-, x, x^+, x_i^-} \left[ l \left( \{f(x)^T(f(x^+) - f(x_i^-))\}_{i \notin I^+} \right) \mid |I^+| < k \right]$$

which always contrasts points from different classes, since it only considers the negative samples that are not from $c^+$. The generalization error is:

$$Gen_M = O \left( \frac{R\sqrt{k}\mathcal{R}_{\mathcal{S}}(\mathcal{F})}{M} + (R^2 + \log k)\sqrt{\frac{\log \frac{1}{\delta}}{M}} \right)$$

where $\mathcal{R}_{\mathcal{S}}(\mathcal{F}) = \mathbb{E}_{\sigma \sim \{\pm 1\}^{(k+2)dM}}[\sup_{f \in \mathcal{F}} \langle \sigma, f_{|\mathcal{S}} \rangle]$, where $f_{|\mathcal{S}} = (f_t(x_j), f_t(x_j^+), f_t(x_{j1}^-), \ldots, f_t(x_{jk}^-))_{j \in [M], t \in [d]}$ .

For $c^+, c_1^-, \ldots, c_k^- \sim \rho^{k+1}$, let $\tau_k = \mathbb{P}[I^+ \neq \emptyset]$ and $\tau_0 = \mathbb{P}[c^+ = c_i^-, \forall i]$. Observe that $\tau_1$, as defined in Section 4.1, is $\mathbb{P}[c^+ = c_1^-]$. Let $p_{\max}(\mathcal{T}) = \max_c \mathcal{D}_{\mathcal{T}}(c)$ and

$$\rho_{\min}^+(\mathcal{T}) = \min_{c \in \mathcal{T}} \mathbb{P}_{c^+, c_i^- \sim \rho^{k+1}}[c^+ = c | Q = \mathcal{T}, I^+ = \emptyset] \tag{21}$$

. In Theorem B.1 we will upper bound the following quantity: $\mathbb{E}_{\mathcal{T} \sim \mathcal{D}} \left[ \frac{\rho_{\min}^+(\mathcal{T})}{p_{\max}(\mathcal{T})} L_{sup}^\mu(\mathcal{T}, \hat{f}) \right]$ ($\mathcal{D}$ was defined in Section 5).

**Theorem B.1.** *Let $\hat{f} \in \arg\min_{f \in \mathcal{F}} \hat{L}_{un}(f)$. With probability at least $1 - \delta$, for all $f \in \mathcal{F}$*

$$\mathbb{E}_{\mathcal{T} \sim \mathcal{D}} \left[ \frac{\rho_{\min}^+(\mathcal{T})}{p_{\max}(\mathcal{T})} L_{sup}^{\mu_{down}}(\mathcal{T}, \hat{f}) \right] \leq \frac{1 - \tau_0}{1 - \tau_k} L_{un}^{\neq}(f) + \frac{c_0 k \tau_1}{1 - \tau_k} s(f) + \frac{1}{1 - \tau_k} Gen_M - \frac{1}{1 - \tau_k} B(f)$$

*where $c_0$ is a constant.*

We now prove the Theorem in 3 steps. We denote by $\mu_c^{\text{uns}} = \mathbb{E}_{x \sim D_c}[f(x)]$ the class mean under the unsupervised distribution, and by $\mu_c^{\text{down}} = \mathbb{E}_{x \sim D_c'}[f(x)]$ the class mean under the downstream (shifted) distribution. The shift is $\delta_c := \mu_c^{\text{uns}} - \mu_c^{\text{down}}$. Thus, $\mu_c^{\text{uns}} = \mu_c^{\text{down}} + \delta_c$. Also, Recall that the sampling procedure for unsupervised data is as follows : sample $c^+, c_1^-, ..., c_k^- \sim \rho^{k+1}$ and then $x, x^+ \sim D_{c^+}^2$ and $x_i^- \sim D_{c_i^-}, i \in [k]$.

**Step 1 (convexity):** By convexity of $\ell$ and Jensen's inequality, we have

$$L_{\text{un}}(\hat{f}) = \mathbb{E}_{\substack{c^+, c_i^- \sim \rho^{k+1} \\ x \sim \mathcal{D}_{c^+}}} \mathbb{E}_{\substack{x_i^- \mathcal{D}_{c_i^-} \\ x^+ \sim \mathcal{D}_{c^+}}} \left[ \ell(\{\hat{f}(x)^\top(\hat{f}(x^+) - \hat{f}(x_i^-))\}_{i=1}^k) \right] \tag{22}$$

$$\geq \mathbb{E}_{\substack{c^+, c_i^- \sim \rho^{k+1} \\ x \sim \mathcal{D}_{c^+}}} \left[ \ell(\{\hat{f}(x)^\top(\mu_{c^+}^{\text{uns}} - \mu_{c_i^-}^{\text{uns}})\}_{i=1}^k) \right] \tag{23}$$

$$= \mathbb{E}_{\substack{c^+, c_i^- \sim \rho^{k+1} \\ x \sim \mathcal{D}_{c^+}}} \left[ \ell(\{\hat{f}(x)^\top(\mu_{c^+}^{\text{down}} - \mu_{c_i^-}^{\text{down}})\}_{i=1}^k + \hat{f}(x)^\top(\delta_{c^+} - \delta_{c_i^-})\}_{i=1}^k) \right]. \tag{24}$$

Applying a first-order Taylor expansion around the unshifted means yields

$$L_{\text{un}}(\hat{f}) \geq \mathbb{E}_{\substack{c^+, c_i^- \sim \rho^{k+1} \\ x \sim \mathcal{D}_{c^+}}} \left[ \ell(\{\hat{f}(x)^\top (\mu_{c^+}^{\text{down}} - \mu_{c_i^-}^{\text{down}})\}_{i=1}^k) + \langle \nabla \ell(\cdot), \{\hat{f}(x)^\top (\delta_{c^+} - \delta_{c_i^-})\}_{i=1}^k \rangle \right]. \tag{25}$$

Define the supremum of the shift-induced contribution:

$$B(f) := \sup_{\delta_c, \delta_c'} \left| \langle \nabla \ell(u), \hat{f}(x)^\top (\delta_c - \delta_{c'}) \rangle \right| \geq 0.$$

**Step 2 (decomposing into supervised tasks)** We now decompose the above quantity to handle repeated classes.

$$\mathbb{E}_{c^+, c_i^- \sim \rho^{k+1}, x \sim D_{c^+}} \left[ \ell\left( \{\hat{f}(x)^T (\mu^{down}_{c^+} - \mu^{down}_{c_i^-})\}_{i=1}^k \right) \right]$$

$$\geq (1 - \tau_k) \mathbb{E}_{c^+, c_i^- \sim \rho^{k+1}, x \sim \mathcal{D}_{c^+}} \left[ \ell\left( \{\hat{f}(x)^T (\mu^{down}_{c^+} - \mu^{down}_{c_i^-})\}_{i=1}^k \right) \Big| I^+ = \emptyset \right]$$

$$+ \tau_k \mathbb{E}_{c^+, c_i^- \sim \rho^{k+1}} [\ell(\underbrace{0, \ldots, 0}_{|I^+| \text{ times}})|I^+ \neq \emptyset] + (1 - \tau_k) B(f)$$

$$\geq (1 - \tau_k) \mathbb{E}_{c^+, c_i^- \sim \rho^{k+1}, x \sim \mathcal{D}_{c^+}} \left[ \ell\left( \{\hat{f}(x)^T (\mu^{down}_{c^+} - \mu^{down}_c)\}_{c \in Q, c \neq c^+} \right) \Big| I^+ = \emptyset \right]$$

$$+ \tau_k \mathbb{E}_{c^+, c_i^- \sim \rho^{k+1}} [\ell_{|I^+|}(\vec{0})|I^+ \neq \emptyset] + (1 - \tau_k) B(f) \tag{24}$$

where $l_t(\vec{0}) = l(0, \ldots, 0)$ ($t$ times). Note that the term $\tau_k B(f) \mathbb{E}[l(0, ..., 0)|I^+ \neq \emptyset]$ will be lined out due to the fact that if classes have a collision, then B(f) will be zero as $\delta$ equals zero.

Recall that in the main paper, sampling $\mathcal{T}$ from $\mathcal{D}$ is defined as sampling the $(k + 1)$-tuple from $\rho^{k+1}$ conditioned on $I^+ = \emptyset$ and setting $\mathcal{T} = Q$. Based on this definition, by the tower property of expectation, we have

$$\mathbb{E}_{c^+, c_i^- \sim \rho^{k+1}, x \sim \mathcal{D}_{c^+}} \left[ \ell\left( \{\hat{f}(x)^T (\mu^{down}_{c^+} - \mu^{down}_c)\}_{c \in Q, c \neq c^+} \right) \Big| I^+ = \emptyset \right]$$

$$= \mathbb{E}_{\mathcal{T} \sim \mathcal{D}_{c^+}, c_i^- \sim \rho^{k+1}, x \sim \mathcal{D}_{c^+}} \left[ \ell\left( \{\hat{f}(x)^T (\mu^{down}_{c^+} - \mu^{down}_c)\}_{c \in Q, c \neq c^+} \right) \Big| Q = \mathcal{T}, I^+ = \emptyset \right]$$

$$= \mathbb{E}_{\mathcal{T} \sim \mathcal{D}_{c^+} \sim \rho^+(\mathcal{T}), x \sim \mathcal{D}_{c^+}} \left[ \ell\left( \{\hat{f}(x)^T (\mu^{down}_{c^+} - \mu^{down}_c)\}_{c \in \mathcal{T}, c \neq c^+} \right) \right] \tag{25}$$

where $\rho^+(\mathcal{T})$ is the distribution of $c^+$ when $(c^+, c_1^-, \ldots, c_k^-)$ are sampled from $\rho^{k+1}$ conditioned on $Q = \mathcal{T}$ and $I^+ = \emptyset$. Recall that $\rho^+_{\min}(\mathcal{T})$ from the theorem's statement is exactly the minimum out of these $|\mathcal{T}|$ probabilities. Now, to lower bound the last quantity with the LHS in the theorem statement, we just need to observe that for all tasks $\mathcal{T}$:

$$\mathbb{E}_{c^+ \sim \rho^+(\mathcal{T}), x \sim \mathcal{D}_{c^+}} \left[ \ell\left( \{\hat{f}(x)^T (\mu^{down}_{c^+} - \mu^{down}_c)\}_{c \in \mathcal{T}, c \neq c^+} \right) \right]$$

$$\geq \frac{\rho^+_{\min}(\mathcal{T})}{p_{\max}(\mathcal{T})} \mathbb{E}_{c^+ \sim \mathcal{D}_{\mathcal{T}}, x \sim \mathcal{D}_{c^+}} \left[ \ell\left( \{\hat{f}(x)^T (\mu^{down}_{c^+} - \mu^{down}_c)\}_{c \in \mathcal{T}, c \neq c^+} \right) \right]$$

$$= \frac{\rho^+_{\min}(\mathcal{T})}{p_{\max}(\mathcal{T})} L^{\mu_{down}}_{sup}(\mathcal{T}, \hat{f}) \tag{26}$$

By combining Equations 23, 24 and 26 we get

$$(1 - \tau_k) \mathbb{E}_{\mathcal{T} \sim \mathcal{D}} \left[ \frac{\rho^+_{\min}(\mathcal{T})}{p_{\max}(\mathcal{T})} L^{\mu_{down}}_{sup}(\mathcal{T}, \hat{f}) \right]$$

$$\leq L_{un}(\hat{f}) - \tau_k \mathbb{E}_{c^+, c_i^- \sim \rho^{k+1}} [l_{|I^+|}(\vec{0})|I^+ \neq \emptyset] - (1 - \tau_k) B(f) \tag{27}$$

Now, by applying Lemma A.1, we bound the generalization error: with probability at least $1 - \delta, \forall f \in \mathcal{F}$

$$L_{un}(\hat{f}) \leq L_{un}(f) + Gen_M \tag{28}$$

**Step 3 ($L_{un}$ decomposition)** Now, we decompose $L_{un}(f)$

$$L_{un}(f) \leq \underset{\substack{c^+, c_i^- \sim \rho^{k+1} \\ x_i^- \sim \mathcal{D}_{c_i^-} \\ x, x^+ \sim \mathcal{D}_{c^+}^2}}{\mathbb{E}} \left[ \ell(\{f(x)^T(f(x^+) - f(x_i^-))\}_{i \notin I^+}) + \ell(\{f(x)^T(f(x^+) - f(x_i^-))\}_{i \in I^+}) \right]$$

$$= \underset{\substack{c^+, c_i^- \sim \rho^{k+1} \\ x, x^+ \sim \mathcal{D}_{c^+}^2 \\ x_i^- \sim \mathcal{D}_{c_i^-} \\ i \notin I^+}}{\mathbb{E}} \left[ \ell(\{f(x)^T(f(x^+) - f(x_i^-))\}_{i \notin I^+}) \right]$$

$$+ \underset{\substack{c^+, c_i^- \sim \rho^{k+1} \\ x, x^+ \sim \mathcal{D}_{c^+}^2 \\ x_i^- \sim \mathcal{D}_{c_i^-} \\ i \in I^+}}{\mathbb{E}} \left[ \ell(\{f(x)^T(f(x^+) - f(x_i^-))\}_{i \in I^+}) \right]$$

$$= (1 - \tau_0) \underset{\substack{c^+, c_i^- \sim \rho^{k+1} \\ x, x^+ \sim \mathcal{D}_{c^+}^2 \\ x_i^- \sim \mathcal{D}_{c_i^-} \\ i \notin I^+}}{\mathbb{E}} \left[ \ell(\{f(x)^T(f(x^+) - f(x_i^-))\}_{i \notin I^+}) \mid |I^+| < k \right]$$

$$+ \tau_k \underset{\substack{c^+, c_i^- \sim \rho^{k+1} \\ x, x^+ \sim \mathcal{D}_{c^+}^2 \\ x_i^- \sim \mathcal{D}_{c_i^-} \\ i \in I^+}}{\mathbb{E}} \left[ \ell(\{f(x)^T(f(x^+) - f(x_i^-))\}_{i \in I^+}) \mid I^+ \neq \emptyset \right] \tag{29}$$

Observe that the first term is exactly $(1 - \tau_0)L_{un}^{\neq} = (f)$. Thus, combining equations 27, 28 and 29 we get

$$(1 - \tau_k)\mathbb{E}_{\mathcal{T} \sim \mathcal{D}} \left[ \frac{\rho_{\min}^+(\mathcal{T})}{p_{\max}(\mathcal{T})} L_{sup}^\mu(\mathcal{T}, \hat{f}) \right] + B(f) \leq (1 - \tau_0)L_{un}^{\neq}(f) + Gen_M \tag{30}$$

$$+ \tau_k \underbrace{\underset{c^+, c_i^- \sim \rho^{k+1}}{\mathbb{E}} \left[ \underset{\substack{x_i^- \sim \mathcal{D}_{c_i^-}, i \in I^+ \\ x, x^+ \sim \mathcal{D}_{c^+}^2}}{\mathbb{E}} [\ell(\{f(x)^T(f(x^+) - f(x_i^-))\}_{i \in I^+})] - \ell_{|I^+|}(\vec{0}) \,\middle|\, I^+ \neq \emptyset \right]}_{\Delta(f)} \tag{31}$$

From the definition of $I^+$, $c_i^- = c^+, \forall i \in I^+$. Thus, from Lemma A.1, we get that

$$\Delta(f) \leq c_0 \mathbb{E}_{c^+, c_i^- \sim \rho^{k+1}} \left[ |I^+|\sqrt{||\Sigma(f, c)||_2} \mathbb{E}_{x \sim \mathcal{D}_c}[||f(x)||] \,\middle|\, I^+ \neq \emptyset \right] \tag{32}$$

for some constant $c_0$. Let $u$ be a distribution over classes with $u(c) = \mathbb{P}_{c^+, c_i^- \sim \rho^{k+1}}[c^+ = c | I^+ \neq \emptyset]$ and it is easy to see that $u(c) \propto \rho(c)(1 - (1 - \rho(c))^k)$. By applying the tower property to Equation (32) we have

$$\Delta(f) \leq c_0 \mathbb{E}_{c \sim u} \left[ \mathbb{E}_{c^+, c_i^- \sim \rho^{k+1}}[|I^+||c^+ = c, I^+ \neq \emptyset] \sqrt{||\Sigma(f, c)||_2} \mathbb{E}_{x \sim \mathcal{D}_c}[||f(x)||] \right]$$

But,

$$\mathbb{E}_{c^+,c_i^- \sim \rho^{k+1}}[|I^+| | c^+ = c, I^+ \neq \emptyset] = \sum_{i=1}^{k} \mathbb{P}_{c^+,c_i^- \sim \rho^{k+1}}[c_i^- = c^+ | c^+ = c, I^+ \neq \emptyset]$$

$$= k\mathbb{P}_{c^+,c_i^- \sim \rho^{k+1}}[c_1^- = c^+ | c^+ = c, I^+ \neq \emptyset]$$

$$= k\frac{\mathbb{P}_{c^+,c_i^- \sim \rho^{k+1}}[c_1^- = c^+ = c]}{\mathbb{P}_{c^+,c_i^- \sim \rho^{k+1}}[c^+ = c, I^+ \neq \emptyset]}$$

$$= k\frac{\rho(c)^2}{\rho(c)(1 - (1 - \rho(c))^k)} = \frac{k\rho(c)}{1 - (1 - \rho(c))^k}$$

Now, using the fact that $\tau_k = 1 - \sum_{c'} \rho(c')(1 - \rho(c'))^k = \sum_{c'} \rho(c')(1 - (1 - \rho(c'))^k)$ and $\tau_1 = \sum_c \rho(c)^2$,

$$\frac{\tau_k}{1 - \tau_k}\Delta(f) \leq \frac{\tau_k}{1 - \tau_k}c_0\mathbb{E}_{c \sim u}\frac{k\rho(c)}{1 - (1 - \rho(c))^k}\sqrt{||\Sigma(f,c)||_2}\mathbb{E}_{x \sim \mathcal{D}_c}[||f(x)||]$$

$$= \frac{c_0 k \tau_k}{1 - \tau_k}\sum_c \rho(c)^2\frac{1}{\sum_{c'} \rho(c')(1 - (1 - \rho(c'))^k)}\sqrt{||\Sigma(f,c)||_2}\mathbb{E}_{x \sim \mathcal{D}_c}[||f(x)||]$$

$$= \frac{c_0 k \tau_1}{1 - \tau_k}\mathbb{E}_{c \sim \nu}[\sqrt{||\Sigma(f,c)||_2}\mathbb{E}_{x \sim \mathcal{D}_c}[||f(x)||]] = \frac{c_0 k \tau_1}{1 - \tau_k}s(f)$$

and we are done.

## C    BIAS DECOMPOSITION FOR DOMAIN GENERALIZATION

**Proposition C.1.** *Consider $\mathcal{C}_{\text{pre}}$ to be the set of latent classes observed meanwhile of pretraining with means $\{\mu_c\}_{c \in \mathcal{C}_{\text{pre}}}$. For any downstream task involving a novel class label $c' \notin \mathcal{C}_{\text{pre}}$ with downstream mean $\mu'_{c'}$, consider $\tilde{\mu}_{c'}$ to be the projection of $\mu'_{c'}$ onto the convex hull of the pretraining means:*

$$\tilde{\mu}_{c'} = \underset{z \in \text{conv}(\{\mu_c\}_{c \in \mathcal{C}_{\text{pre}}})}{\arg\min} \|z - \mu'_{c'}\|.$$

*We formulate the geometric generalization cost as the residual norm $\|r_{c'}\| = \|\tilde{\mu}_{c'} - \mu'_{c'}\|$. Assuming that the loss $\ell$ is $L$-Lipschitz and representations are bounded by $R$, the bias contribution for class $c'$ satisfies:*

$$B_{c'}(\hat{f}) \leq B_{\text{in-dist}}(\hat{f}, \tilde{\mu}_{c'}) + 2LR \cdot \|r_{c'}\|,$$

*where $B_{\text{in-dist}}$ represents the shift bias relative to the closest valid pretraining proxy $\tilde{\mu}_{c'}$, and the second term explicitly penalizes the distance to the pretraining manifold.*

## D    ADDITIONAL EMPIRICAL ANALYSIS

The classification results on the PACS dataset using a ResNet-50 encoder are summarized in Table 1. Consistent with the challenging nature of Domain Generalization on PACS Li et al. (2017), we observe a notable performance drop when moving from average binary tasks (AVG-2) to the full classification task (AVG-7). However, several key patterns emerge that validate our theoretical framework.

First, the unsupervised performance is remarkably close to the supervised baseline across most domains. This empirical proximity supports our theoretical derivation that the unsupervised contrastive loss $L_{un}$ acts as a surrogate for the supervised loss $L_{sup}$, particularly in the binary classification setting (AVG-2).

Second, regarding the classifier efficiency, the mean classifier ($\mu$) proves to be a robust estimator, performing comparably to the trained linear layer (TR). Furthermore, the representations exhibit

| | | SUPERVISED | | | UNSUPERVISED | | | SHIFT MAG. |
|---|---|---|---|---|---|---|---|---|
| | | TR | $\mu$ | $\mu$-5 | TR | $\mu$ | $\mu$-5 | $\delta$ |
| PHOTO | AVG-2 | 83.1 | 79.3 | 78.4 | 81.5 | 79.0 | 77.2 | 3.43 |
| | AVG-7 | 55.5 | 50.4 | 49.2 | 31.3 | 45.8 | 30.8 | |
| ART PAINTING | AVG-2 | 78.2 | 76.7 | 76.2 | 76.6 | 75.8 | 74.9 | 4.65 |
| | AVG-7 | 37.3 | 35.1 | 27.8 | 25.1 | 24.4 | 23.6 | |
| CARTOON | AVG-2 | 74.4 | 74.2 | 73.9 | 72.5 | 72.2 | 71.7 | 5.50 |
| | AVG-7 | 35.1 | 34.4 | 33.6 | 50.0 | 45.3 | 36.7 | |
| SKETCH | AVG-2 | 69.9 | 69.7 | 68.6 | 66.4 | 65.9 | 65.4 | 7.72 |
| | AVG-7 | 30.3 | 30.7 | 28.8 | 26.0 | 24.4 | 26.1 | |

Table 1: Classification results on the PACS dataset with Shift Magnitude.

strong concentration properties: the mean classifier estimated from only 5 labeled samples ($\mu$-5) achieves performance very close to the full sample mean ($\mu$). This suggests that the learned representation space is well-structured even with minimal supervision.

Finally, and most importantly, the results validate the role of the bias term $B(f)$ in our generalization bounds. We observe a strong inverse correlation between the **Shift Magnitude** ($\delta$) and downstream accuracy. The "Photo" domain, which exhibits the smallest distributional shift ($\delta = 3.43$), retains the highest performance stability. Conversely, the "Sketch" domain, which represents a severe domain shift with the largest magnitude ($\delta = 7.72$), suffers the lowest accuracy. This confirms that $\delta$ effectively captures the representational misalignment and the resulting transfer difficulty predicted by our theory.

