# OpenReview forum: "Revisiting Theory of Contrastive Learning for Domain Generalization"
_ICLR.cc/2026/Conference — Submitted to ICLR 2026_

### Official Review · Reviewer_pRMN · 2025-10-28

**Soundness:** 2
**Presentation:** 3
**Contribution:** 2
**Rating:** 4
**Confidence:** 4

**Summary:**

The paper proposes a novel and *realistic* generalization bounds for contrastive learning under both domain shift and domain generalization (distinguishing the two is one of the paper's main ideas). The authors show that downstream performance depends on the statistical discrepancy between pretraining and downstream distributions through an explicit bias term, thereby providing guarantees in both scenarios. In other words, representations learned through contrastive learning transfer well when downstream class means remain close to the pretraining latent means.

**Strengths:**

Thank you for your work, I have truly enjoyed reading the paper. Below, I have listed the strengths of this paper.

- Realistic Assumptions in Contrastive Learning (CL) X Domain Generalization (DG): First and foremost, it is worth mentioning that the paper addresses an important, yet often overlooked issue in DG: the in-distribution assumption. I believe this is the largest, and most important contribution of this paper.
- Technical Correctness: The derivations proposed in the paper are sound and mathematically consistent.
- Presentation: The paper is well-written, and the motivations are very clear and easy to understand.

**Weaknesses:**

While the paper has some strengths, it also has critical weaknesses.

- Lack of empirical grounding: While the theoretical analysis is sound, there is a visible lack of experimental results. While I acknowledge that this paper's contributions derive in its theoretical aspects, the paper would benefit from even small-scale experiments. Frankly, this is my largest concern regarding the paper. Please refer to the following Questions section for potential experiments.
- Partially novel, yet incremental contribution: The gap between CL and DG is a topic deeply studied in the last few years. While the paper extends Saunshi et al's [1] framework to a more realistic setting, the main observations (e.g., CL provably degrades under realistic distribution shifts and thus better sampling/alignment is needed) are already present.

***

### References
[1] Saunshi et al., A theoretical analysis of contrastive unsupervised representation learning, ICML, 2019

**Questions:**

Overall, I view this paper to be theoretically solid and promising, but there are some questions that *must* be addressed.


- I believe adding some empirical analysis would strengthen the paper substantially. For instance:
    - Estimate mean shifts in distributions and correlate them with transfer accuracy.
    - Could you provide some experiments, at least the ones performed in [1]? The lack of empirical results fades the theoretical contributions of this paper, especially in a venue like ICLR, where empirical results are also expected.
    - Validate the predicted role of the intra-class variance $s(f)$
    - How does increasing $k$ trade off with collision driven penalities?
- Correct me if I'm wrong, but the guarantees focus on mean classifiers. While the paper assumes the use of mean classifiers, in reality, fine-tuned heads are used. Could you elaborate how the paper's conclusions could be extended to scenarios where fine-tuned classifiers are used? e.g., Are there stability assumptions under which the conclusions still go through?


***

### References
[1] Saunshi et al., A theoretical analysis of contrastive unsupervised representation learning, ICML, 2019

---

> ### Author Response · Authors · 2025-11-23
> **Response to Reviewer pRMN Part 1/2**
>
> We thank the reviewer for the thoughtful and encouraging assessment.
>
> ### W1, Q1: Empirical Experiments
> >* … , the paper would benefit from even small-scale experiments
> >* Q1.1: Estimate mean shifts in distributions and correlate them with transfer accuracy.
>
> Thank you for raising this point. We now provide precisely this analysis. For CIFAR--10 $\rightarrow$ CIFAR--10--C, the estimated shift magnitudes correlate strongly (approximately linearly) with the downstream accuracy drop, validating the predictive structure of Theorems provided in the paper.
>
> We draw your attention to the new version of our paper, specifically the new Section 7. Empirical Analysis, Figure 1 .
>
>
> >* Q1.2: Could you provide some experiments, at least the ones performed in [1]? The lack of empirical results fades the theoretical contributions of this paper, especially in a venue like ICLR, where empirical results are also expected.
>
> We appreciate this suggestion. Our new experiments validate the structure of the $shift-dependent$ bound; we opted for corruption-based shifts (CIFAR--10-C) rather than repeating all experiments from [1], but we will clarify this rationale in the revision.
>
>
> >* Q1.3: How does increasing trade off with collision driven penalities?
>
> Based on our theoretical analysis (Theorem 5.1 and the definitions in Appendix B), the "trade-off" arises due to increasing the number of negatives $k$ increases the probability of **false negatives (collisions)**, which loosens your generalization guarantee.
>
>
> ### W2: incremental contribution:
> >* comparison with CL provably degrades under realistic distribution shifts and thus better sampling/alignment is needed
>
> Although our analysis builds on Saunshi et al. (2019), our work introduces essential new components required to handle domain shift and domain generalization:
> 1. **Novel decomposition for downstream shift.**   Unlike Saunshi et al., who assume identical pretraining and downstream distributions, our analysis requires   **reparameterizing with shifted downstream means $\mu'_c$** and introducing the **explicit shift vector $\delta_c$**.  This alters the structure of the supervised loss fundamentally (Lemma 4.3).
> 2. **New bias term B(f) with explicit shift dependence.**  We introduce the term $B(f)≈E\[ℓ′(f(x)⊤(μ_c−μ_c′​))f(x)⊤(\delta_c​−\delta_c′​)\]$  which captures **first-order representational misalignment** between upstream and downstream domains.  This quantity does not appear in prior contrastive-learning theory and provides a principled explanation for performance degradation under distribution shift.
> 3. **Integration of downstream classes.**  Our framework accommodates downstream classes not observed during pretraining by bounding their means of the pretraining class means $\mu_c$. Previous theoretical analyses do not address label-space expansion.
> 4. **Estimating the shift in practice** Although $\mu_c$ is unobserved, $\delta_c$ can be approximated using simple heuristics such as averaging positive-pair embeddings, clustering representations into pseudo-classes, or exploiting semantic grouping in large pretrained models. These approximations make the shift-dependent bounds practically usable.
> 5. **Generalization to \( k \) negatives under shift.**  Appendix B (Theorem B.1) extends the bounds to the multi-negative setting **in the presence of distribution shift**,  requiring a new treatment of class-collision patterns and conditional sampling distributions.  This represents a substantive technical extension beyond earlier work.
>
>
> ### Q2: extension of our work on fine-tuning
> >* Q2.1: Correct me if I'm wrong, but the guarantees focus on mean classifiers. While the paper assumes the use of mean classifiers, in reality, fine-tuned heads are used. Could you elaborate how the paper's conclusions could be extended to scenarios where fine-tuned classifiers are used?
>
> The reviewer is correct that our formal guarantees are stated for linear mean classifiers, consistent with the latent-class framework of [1]. However, the theoretical conclusions extend naturally to fine-tuned linear heads under mild stability assumptions: If the downstream head $w^*$ satisfies
> $ \|w^\star - w^{\mathrm{mean}}\| \le \alpha \cdot s(f) \quad\text{and}\quad \|w^\star\| \le R,$
> then the downstream loss differs from the mean-classifier loss by at most $O(\alpha\, s(f))$, which vanishes when intra-class variance is small.
> Intuitively, when $s(f)$ is small (as is typical for modern CL representations), fine-tuned classifiers remain close to the mean classifier and inherit the same shift-dependent generalization behavior.
> We will add a discussion paragraph clarifying this point and outlining how such stability arguments can be formalized.

---

> > ### Author Response · Authors · 2025-11-23
> > **Response to Reviewer pRMN Part 2/2**
> >
> > >* Q2.2:  Are there stability assumptions under which the conclusions still go through?
> >
> > We agree that bridging the gap between the analyzed mean classifier ($W_{\mathrm{mean}}$) and a fine-tuned linear classifier $W^\ast$ is crucial. A natural way to extend our guarantees to fine-tuned classifiers is through a simple
> > \emph{stability} assumption on the downstream head. In practice, the fine-tuned linear classifier
> > $W^\ast$ typically remains close to the mean classifier $W_{\mathrm{mean}}$ because the encoder
> > already produces well-clustered representations. Under the mild condition
> > \[
> > $\|w^* - w^{\mathrm{mean}}\| \le \alpha\, s(f)$
> > $\quad\text{and}\quad$
> > $\|W^\ast\| \le R$
> > \]
> > The downstream loss differs from the mean-classifier loss by at most $O(\alpha\, s(f))$.
> > Thus, when the intra-class variance $s(f)$ is small---as is often the case for modern
> > contrastive encoders---the fine-tuned head inherits essentially the same shift-dependent
> > generalization behavior.

---

> ### Comment · Reviewer_pRMN · 2025-11-25
>
> I thank the authors for their detailed response and revision, including the new empirical section, which studies the role of mean shift and addresses several of my technical concerns. However, my primary concern about the lack of empirical grounding remains: to substantiate the claim of the theory, I believe the paper should at least reproduce the core experiments of prior work (such as Saunshi et al.) and include results on standard DG benchmarks (e.g., DomainBed) if it aims to position itself as a work in DG. In its current form, even though the theoretical analysis is sound and well-presented, the empirical analysis is too limited for me to update my overall assessment. I believe the paper has a strong potential to develop into a much stronger paper if it includes standard experiments shared across the DG literature. I am keeping my original score.

---

> > ### Author Response · Authors · 2025-12-01
> > **Response to Reviewer pRMN**
> >
> > We thank the reviewer for suggesting the additional empirical study and comparison with prior work (e.g., Saunshi et al.) on standard DG benchmarks such as DomainBed. We have included these new results in Appendix D (page 18) of the revised manuscript. Notably, we observe a strong inverse correlation between the Shift Magnitude ($\delta$) and downstream accuracy.

---

### Official Review · Reviewer_5uF7 · 2025-10-30

**Soundness:** 3
**Presentation:** 2
**Contribution:** 3
**Rating:** 4
**Confidence:** 4

**Summary:**

This paper extends the theoretical framework of contrastive learning (CL), originally formalized by Saunshi et al. (2019), to explicitly account for domain shift and domain generalization. Classical theory assumes that downstream tasks share the same latent class distribution as pretraining, an assumption violated in realworld transfer scenarios. The authors introduce a generalization bound for contrastive representations that includes a bias term $B(f)$, quantifying representational misalignment between pretraining and downstream distributions.


They derive bounds for the supervised downstream loss under two settings: (i) shifted distributions within the same label space and (ii) novel downstream classes outside the pretraining latent set. The paper also extends the theoretical analysis to multiple negative samples, bounding how class collisions and intra-class variance affect guarantees. Additional analysis is provided under Lipschitz continuity and sub-Gaussian assumptions for the encoder, yielding interpretable upper bounds that scale linearly with the shift magnitude $\epsilon$. The authors conclude by situating their work as a unifying theoretical account of contrastive learning robustness and transferability.

**Strengths:**

Overall, the paper demonstrates theoretical soundness. Its central claim that distributional mismatches introduce a quantifiable bias term in the generalization bound is well supported. The proofs are largely self-contained and presented with sufficient rigor, and the results hold consistently across multiple loss functions (hinge and logistic), which strengthens their generality. No major logical flaws or unjustified leaps were found. Other notable points are stated below.

--Ambitious attempt to bridge contrastive-learning theory with domain-shift analysis.

--The derived bias bounds (scaling with shift magnitude $\epsilon$) are interpretable and could guide empirical model evaluation

--Useful decomposition of the unsupervised loss into shifted-mean and intra-class-variance terms; conceptually interesting direction for future work.

**Weaknesses:**

The work relies on idealized assumptions, such as access to latent class means and uniform downstream sampling, which may not hold in realistic settings as they agree in the end of Section 4.


Lack of literature review: Although the paper builds directly upon and extends the framework of Saunshi et al. (2019), the authors do not clearly summarize that prior work or provide a comprehensive comparison with its results. This omission weakens the contextual grounding of their contribution.


The abstract promises guarantees for new downstream classes, but no theorem contains a term accounting for unseen classes. All bounds assume shared label spaces with bounded mean shifts, i.e., domain shift only.


The appendix introduces $\tau_k, \tau_0, \rho_{\min}^+, p_{\max}$ but not provide explicit scaling or normalization, making it impossible to check consistency with Theorem 5.1’s informal statement.


Incomplete presentation: Several presentation issues reduce the clarity of the work and make it hard to fully evaluate.  Some examples are noted below.

- Missing definitions for key notations:  $W^\mu$ in line 114, $R$ and $\Re_S(\mathcal{F})$ in line 168

- The inequality on line 720 extends beyond the visible text boundary, leaving it unreadable

 - An incomplete sentence appears around lines 237–238

**Questions:**

1. In Section 4.2, both similar pairs and negative examples can originate from either the same or different classes. So what specific criterion is used to define similar and negative examples in this context? Does “similar” refer to an augmentation of the anchor sample, or is there another underlying assumption?

2. The paper mentions measuring the discrepancy of novel downstream class means by their distance to the convex hull of pretraining means. However, this concept does not appear explicitly in any theorem. Could you clarify where this notion is formally incorporated or demonstrate how it can be derived within your framework?

3. Please provide a clearer comparison between your theoretical results and those of Saunshi et al. (2019). Specifically, what key assumptions or bounds have been relaxed or strengthened relative to their work?

4. Could your theoretical framework be extended to contrastive multimodal pretraining (e.g., CLIP), where the distributional shifts occur across different modalities rather than domains?

---

> ### Author Response · Authors · 2025-11-23
> **Response to Reviewer 5uF7 Part 1/2**
>
> We thank the reviewer for the useful comments and questions. We address the weaknesses (W) and questions (Q) as follows:
>
> ### W1: clarification on our assumption
>
> We agree that our assumption for $\mu$, which we mentioned at the end of Section 6 is somehow idealized. We state that $\mu$’s are not directly observable due to the inherent characteristic of contrastive learning, but we explicitly discuss how to relax the latent-mean assumption in practice in Section 6 under **Estimating the shift in practice**: $\mu_c$ and hence $\delta_c$ can be approximated by (i) averaging positive pairs under augmentations, (ii) clustering embeddings into pseudo-classes, and (iii) exploiting the strong semantic grouping of large pretrained models.
> These approximations make the shift-dependent bias term $B(f)$ empirically measurable.
>
> For the uniform sampling of downstream classes assumption, this assumption is just used for the theorems in the main paper and for specific cases, to be precise, we used this assumption in 3 places.
> * (1) At the end of page 2, which we haven’t proved or stated any theorem or lemma, and are just using it for simplicity and a typical case.
> * (2) For proving Theorem 4.1 and Lemma 4.3, when splitting two cases $c^+ = c^-$ and $c^+ \neq c^-$, which we proof the general distributions in Theorem B.1, appendix.
> * (3) For Theorem 5.1, which we just stating the value of coefficients under this assumption to giving better sense about them.
>
> We also proof the general distributions in Theorem B.1 Appendix.
> We can add footnote where we use this uniform sampling and stating that the general distribution is proved in appendix.
>
> ### W2.1, Q3: comparison with Saunshi et al.:
>
> Conceptually, our framework reduces to Saunshi et al. when there is no distribution shift ($\delta_c = 0$ for all $c$) and no novel downstream classes.
>
> Relative to their work, we:
> (a) incorporate an explicit downstream bias term $B(f)$ that depends on the shift vectors $\delta_c$ and recovers their in-distribution bound when $\delta_c=0$ (Theorem 4.1, Lemma 4.3, Theorem 4.5);
> (b) extend the class-collision/intra-class-variance analysis to shifted distributions and to $k>1$ negatives (Theorem 5.1 / Theorem A.4);
> and (c) provide Lipschitz and sub-Gaussian upper bounds for $B(f)$ (Section 6), which are absent in the original framework.
>
> ### W2.2: literature review.:
> Since Reviewer ZTZj also raised this concern, here is our answer: We have now expanded more details in our text (Introduction) to explicitly position our contributions within these two lines of literature:
> 1. **Provable guarantees under distribution shift**  Existing OOD/self-supervised generalization theory typically addresses covariate or label shift in supervised settings or assumes access to labeled auxiliary domains. However, none of them provide a latent-class decomposition with explicit shift vectors $\delta_c$  or a downstream bias term $B(f)$.
> 2. **Foundation-model transferability**.  Recent analyses primarily focus on scaling laws and linear mode connectivity [2], or logit geometry [3]. Our contribution is complementary: we provide explicit first-order misalignment bounds relating encoder statistics to transfer degradation. We can add a paragraph explaining how $\delta_c$ and $B(f)$ provide a mechanism-level explanation for phenomena observed empirically in foundation models (e.g., robustness to small distribution shift due to concentrated representations).
>
> ### W3:  guarantees for new downstream classes
> >* The abstract promises guarantees for new downstream classes, but no theorem contains a term accounting for unseen classes. All bounds assume shared label spaces with bounded mean shifts, i.e., domain shift only.The appendix introduces $\tau_k$, $\tau_0$, $\rho^+$, $p_m$, but not provide explicit scaling or normalization, making it impossible to check consistency with Theorem 5.1’s informal statement.
>
> We clarify our formulation as follows:
> **Guarantees for Unseen Classes:** We model Domain Generalization *geometrically* rather than via label matching. As mentioned in Lines 150--152, for a novel downstream class $c'$ not present during pretraining, the ''shift'' vector $\delta$ quantifies the distance between the downstream mean $\mu'_{c'}$ and the **convex hull of pretraining means**. Since representations are norm-bounded ($\|f(x)\| \le R$), this shift is always finite. Therefore, the bound remains valid, though the bias term $B(\hat{f})$ naturally increases to reflect the difficulty of the novel domain.
>
> *Consistency of Theorem 5.1:* We confirm that the Appendix is consistent with the informal statement. Equations on page 18 in Appendix B derive a coefficient proportional to $k\tau_1$. Under a uniform distribution assumption where the collision probability is $\tau_1 \approx 1/|\mathcal{C}|$ (and assuming $k \ll |\mathcal{C}|$), this term simplifies directly to $\beta \approx k/|\mathcal{C}|$. We will add this explicit derivation to the final Appendix.

---

> ### Author Response · Authors · 2025-11-23
> **Response to Reviewer 5uF7 Part 2/2**
>
> ### W4: typos and incomplete presentation
>
> We apologize for the incomplete representation. Now we have revised our manuscript according to your feedback and other reviewers and fixed all grammar issues and typos.
>
> ### Q1: Negative and Positive pairs:
> >* In Section 4.2, both similar pairs and negative examples can originate from either the same or different classes. So what specific criterion is used to define similar and negative examples in this context? Does “similar” refer to an augmentation of the anchor sample, or is there another underlying assumption?
>
> Yes, we confirm that **similar** refers to an augmentation of data, the samples that will be generated from a specific sample to be defined as a positive pair in our unsupervised task (loss). While negative samples refer to those that are drawn independently from the marginal data distribution (i.e., any other random sample in the minibatch)
> The discussion in Section 4.2 regarding samples originating from ''same or different classes'' refers to the latent ground-truth labels. While the algorithm treats any random sample as a negative, our theoretical analysis accounts for the probability that a random negative may coincidentally belong to the same latent class as the anchor (a false negative or ''class collision'').
>
>
> ### Q2: Measuring the discrepancy of novel downstream class means :
> >* Could you clarify where this notion is formally incorporated or demonstrate how it can be derived within your framework?
>
> We have now added a new Proposition C.1, which we have placed in the appendix (Page 18). This proposition formally decomposes the shift into a component within the pretraining hull and a residual term representing the geometric distance to the novel domain. Unfortunately, we could not place it here(OpenReview) due to formatting issues.
>
>
> ### Q4: extension to contrastive multimodal pretraining:
> > * Could your theoretical framework be extended to contrastive multimodal pretraining (e.g., CLIP), where the distributional shifts occur across different modalities rather than domains?
>
> Our theoretical framework is formulated at the level of representations and loss, and does not assume a specific input modality.
> In a multimodal CL setting such as CLIP, one typically has paired data $(x^{(\mathrm{img})}, x^{(\mathrm{text})})$ drawn from a joint latent class, with encoders $f_{\mathrm{img}}$ and $f_{\mathrm{text}}$ trained using a contrastive loss across modalities.
> Conceptually, our analysis can be extended by:
> (i) defining latent classes over paired data and class-conditional distributions $D_c$ on $(x^{(\mathrm{img})}, x^{(\mathrm{text})})$;
> (ii) treating the joint representation (or the image/text representations separately) as $f(x)$ in our framework; and
> (iii) modeling distribution shift as changes in the joint or marginal class-conditional distributions across modalities.
> The resulting bounds would again involve shift vectors between pretraining and downstream means in representation space, and the same type of bias term $B(f)$.
> A full multimodal treatment would require additional notation and is beyond the current paper’s scope, but our framework is compatible with such an extension, which we will mention explicitly as a direction for future work.

---

### Official Review · Reviewer_ZTZj · 2025-11-01

**Soundness:** 2
**Presentation:** 2
**Contribution:** 2
**Rating:** 2
**Confidence:** 3

**Summary:**

This paper extends the theoretical framework of contrastive learning to account for distribution shift and domain generalization. Building upon Saunshi et al. (2019)'s latent class model, the authors introduce generalization bounds that explicitly handle: 1. Domain shift where downstream class distributions differ from pretraining; 2. Domain generalization where downstream tasks involve novel label spaces. The key contribution is a bias term that quantifies representational misalignment between pretraining and downstream distributions, bounded under various assumptions.

**Strengths:**

1. Well-motivated extension: The paper addresses a genuine gap in contrastive learning theory.
2. Clean mathematical framework: The paper provides an elegant way to quantify distribution mismatch. The first-order Taylor expansion approach in Lemma 4.3 is technically sound.

**Weaknesses:**

1. Limited practical impact: No experiments showing whether minimizing the theoretical quantities leads to better downstream performance
2. Incremental technical contribution: The proof technique is a straightforward extension of Saunshi et al. (2019)

**Questions:**

1. Tightness: Can you provide any evidence (theoretical or empirical) that your bounds are not vacuous?
2. Can you provide more experiments to demonstrate the effectiveness of your theory?
3. Connection to recent work: How does your framework relate to:
- Provable guarantees for self-supervised learning under distribution shift ?
- The recent analysis of foundation models' transferability?

---

> ### Author Response · Authors · 2025-11-23
> **Response to Reviewer ZTZj Part 1/2**
>
> We thank the reviewer for the constructive comments and feedback. We address the weaknesses (W) and questions (Q) as follows:
>
> ### W1, Q2: Limited practical impact:
> > * Can you provide more experiments to demonstrate the effectiveness of your theory?
>
> We performed an empirical study considering an encoder (ResNet-34) pretrained contrastively on CIFAR-10 and evaluated its transferability on CIFAR-10-C. For each class, we estimated both (i) the latent mean embeddings from the pretrained encoder and (ii) the shifted downstream means in CIFAR-10-C, enabling us to compute empirical approximations of the shift vectors $\hat\delta_c$. Consistent with our theoretical predictions, we observed:
> * (i) The empirical class-mean shift $\|\delta_c\|$ increases monotonically with corruption severity (predicted by Theorem 4.1).
> * (ii) The downstream classification accuracy decreases linearly with the estimated magnitude of $\delta$ which is the main term in $B(f)≈E\[ℓ′(f(x)⊤(μ_c−μ_c′​))f(x)⊤(\delta_c​−\delta_c′​)\]$ exactly as envisioned by Theorem 4.1 and the linear upper bounds in Section 6.
>
> These results directly support our central hypothesis that distribution-shift-induced performance drop is governed by first-order misalignment in class means, and that the theoretical bias term $B(f)$ captures this behavior.
> Thank you for raising this point, and we have a new section (Section 7: Empirical Results) and included all details in the revised manuscript (at the end of pages 8, 9, and 10)
>
> ### W2: Incremental technical contribution:
>
> Although our analysis builds on Saunshi et al. (2019), our work introduces essential new components required to handle domain shift and domain generalization:
> 1. **Novel decomposition for downstream shift.**   Unlike Saunshi et al., who assume identical pretraining and downstream distributions, our analysis requires   **reparameterizing with shifted downstream means $\mu'_c$** and introducing the **explicit shift vector $\delta_c$**.  This alters the structure of the supervised loss fundamentally (Lemma 4.3).
> 2. **New bias term $B(f)$ with explicit shift dependence.**  We introduce the term $B(f)≈E\[ℓ′(f(x)⊤(μ_c−μ_c′​))f(x)⊤(\delta_c​−\delta_c′​)\]$,  which captures **first-order representational misalignment** between upstream and downstream domains.  This quantity does not appear in prior contrastive-learning theory and provides a principled explanation for performance degradation under distribution shift.
> 3. **Integration of downstream classes.**  Our framework accommodates downstream classes not observed during pretraining by bounding their means of the pretraining class means *$\{\mu_c\}$*. Previous theoretical analyses do not address label-space expansion.
> 4. **Estimating the shift in practice** Although $\mu_c$ is unobserved, $\delta_c$ can be approximated using simple heuristics such as averaging positive-pair embeddings, clustering representations into pseudo-classes, or exploiting semantic grouping in large pretrained models. These approximations make the shift-dependent bounds practically usable.
> 5. **Generalization to \( k \) negatives under shift.**  Appendix B (Theorem B.1) extends the bounds to the multi-negative setting **in the presence of distribution shift**,  requiring a new treatment of class-collision patterns and conditional sampling distributions.  This represents a substantive technical extension beyond earlier work.
>
> ### Q1: Tightness:
> > * Can you provide any evidence (theoretical or empirical) that your bounds are not vacuous?
>
> We performed an empirical study as mentioned above (Section 7.) to ensure the role of $\delta$ and mainly, $B(f)$, and we observed that with increasing magnitude of the shift vector $\delta$, the accuracy of the downstream task drops, validating our theorems and ideas.
> Besides this empirical result, we know that $B(f)$ is bounded under various settings as highlighted in Section 6.

---

> ### Author Response · Authors · 2025-11-23
> **Response to Reviewer ZTZj Part 2/2**
>
> ### Q3: Connection to related works:
> >* How does your framework connected to [1] and [2,3]
> We have now expanded more details in our text (Introduction) to explicitly position our contributions within these two lines of literature:
>
> **Provable guarantees under distribution shift [1]**.  Existing OOD/self-supervised generalization theory typically addresses covariate or label shift in *supervised* settings or assumes access to labeled auxiliary domains. However, none of them provide a latent-class decomposition with explicit shift vectors $\delta_c$  or a downstream bias term $B(f)$.
>
> **Foundation-model transferability**.  Recent analyses primarily focus on scaling laws and linear mode connectivity [2], or logit geometry [3]. Our contribution is complementary: we provide explicit *first-order misalignment bounds* relating encoder statistics to transfer degradation. We can add a paragraph explaining how $\delta_c$ and $B(f)$ provide a mechanism-level explanation for phenomena observed empirically in foundation models (e.g., robustness to small distribution shift due to concentrated representations).
>
> References:
>
> [1]https://proceedings.neurips.cc/paper/2021/file/27debb435021eb68b3965290b5e24c49-Paper.pdf
>
> [2]https://dl.acm.org/doi/10.1145/3696410.3714801
>
> [3]https://arxiv.org/html/2503.09363v1

---

### Comment · Reviewer_ZTZj · 2025-11-24
**Official Comments**

I appreciate  authors' reply.

After reviewing all reviewers' comments and author's reply, I decide to keep my original score.

---

> ### Author Response · Authors · 2025-11-24
> **Reply to Reviewer ZTZj**
>
> Dear Reviewer ZTZj,
>
> We kindly request clarification regarding the specific concerns you believe remain unresolved. Could you please confirm whether you have reviewed the revised version of our manuscript?  If you have any further concerns or suggestions, please do not hesitate to let us know.
>
> Thank you,
> Authors

---

### Author Response · Authors · 2025-12-01
**Key updates and changes on the revised manuscript**

Dear Area Chair and Senior Area Chair,

Contrastive learning is the cornerstone of modern self-supervised learning and foundation models. However, despite its widespread adoption, theoretical understanding of its behavior—particularly under distribution shifts and domain generalization—remains significantly understudied compared to empirical advancements. We believe the ICLR community will benefit from a rigorous framework that explains why and when these representations transfer successfully.

We thank the reviewers for their constructive feedback and their time. In response to the reviews (specifically regarding empirical validation and theoretical clarifications), we have made the following updates to the manuscript:

* New Empirical Validation: Addressing the main request from all reviewers, we added experiments and considered two scenarios (See Section 7, page 8). We demonstrate a strong linear correlation between our theoretical shift magnitude, $\delta$, and downstream accuracy drops. This confirms that our derived bias term $B(f)$ is not just a theoretical artifact, but a predictive measure of transferability in real-world shifts.

* Formalizing Domain Generalization (Appendix C): We addressed concerns regarding "unseen classes" by adding a formal Proposition (See Proposition C.1, in page 18 ). This decomposes the shift for novel classes into a projection onto the convex hull of pretraining means plus a geometric residual, rigorously grounding our claims on Domain Generalization.

* Additional Empirical Study and Comparison with Prior Work (Saunshi et al.) on Standard DG Benchmarks (e.g., DomainBed) as requested in 25of Nov. by Reviewer pRMN: The new results are now in a new section in the Appendix (See Section D , page 18), and we observe a strong inverse correlation between the *Shift Magnitude* ($\delta$) and downstream accuracy.

We believe these revisions address the major and minor concerns regarding practical applicability and theoretical rigor, warranting a re-evaluation of the paper.

Thank you for your consideration,

The Authors

---

### Meta-Review · Area_Chair_pS4F · 2026-01-15

**Summary:**

The paper extends the latent-class theory of contrastive learning to account for distribution shift and domain generalization by introducing a bias terms to extend existing results.  Reviewers found the theory sound and well motivated, and the rebuttal added empirical results to support the theory statement. However, concerns remain that the contribution is relatively incremental, and the validity of the assumptions.

**Reviewer Concerns:**

Addressed: empirical results, comparison with Saunshi et al, multi-negative setting, shift estimation

Outstanding: Unseen classes, assumption validity and theory presentation

**Reviewer Scores:**

5uF7 might increase by 1, the other two expressed to keep the score.

---

### Decision · Program_Chairs · 2026-01-26

Reject